# A consistent map in the medial entorhinal cortex supports spatial memory

Taylor J. Malone [1,7], Nai-Wen Tien [1,5,7], Yan Ma[1,7], Lian Cui[1], Shangru Lyu[1], Garret Wang[1], Duc Nguyen[1,6], Kai Zhang[1,2], Maxym V. Myroshnychenko[3], Jean Tyan [1], Joshua A. Gordon [3,4], David A. Kupferschmidt [3] & Yi Gu[1] ✉

The medial entorhinal cortex (MEC) is hypothesized to function as a cognitive map for memory-guided navigation. How this map develops during learning and influences memory remains unclear. By imaging MEC calcium dynamics while mice successfully learned a novel virtual environment over ten days, we discovered that the dynamics gradually became more spatially consistent and then stabilized. Additionally, grid cells in the MEC not only exhibited improved spatial tuning consistency, but also maintained stable phase relationships, suggesting a network mechanism involving synaptic plasticity and rigid recurrent connectivity to shape grid cell activity during learning. Increased c-Fos expression in the MEC in novel environments further supports the induction of synaptic plasticity. Unsuccessful learning lacked these activity features, indicating that a consistent map is specific for effective spatial memory. Finally, optogenetically disrupting spatial consistency of the map impaired memory-guided navigation in a well-learned environment. Thus, we demonstrate that the establishment of a spatially consistent MEC map across learning both correlates with, and is necessary for, successful spatial memory.

The ability to form a memory of an environment and use the memory to guide future navigation is one of the most fundamental brain functions. This ability relies on the hippocampal-entorhinal circuit, which is hypothesized to construct a cognitive map representing the spatial layout of an environment[1,2]. The medial entorhinal cortex (MEC) is a major component of the map. MEC dysfunction leads to spatial memory deficits in animals and humans and is associated with impaired spatial cognition in Alzheimer's disease[3,4]. The MEC also contains navigation-related cell types, such as grid cells, which fire in a triangular lattice in an open arena[5], and other cell types with activity patterns representing animals' head direction, speed, and environmental borders and landmarks during navigation[6].

Despite the pivotal role of the MEC in spatial memory and its diverse neural activity during navigation, little is known about how MEC activity supports spatial learning and memory. Answering this question requires studies to demonstrate the association between MEC activity features with successful and unsuccessful spatial learning and the necessity of these activity features for effective spatial memory. However, such studies have never been conducted. Although electrophysiological evidence suggests experience-dependent changes in MEC activity stability[5] and patterns[7–11], and stable MEC activity in familiar environments within the same day[12], none of the MEC activity features has been causally associated with spatial learning outcomes. Additionally, these studies suffered from the limitations of electrophysiology, which can only track the activity of tens of MEC neurons over several days[7,8,11] or at best, tens to hundreds of neurons within the same day[9,10,12], and therefore, were not ideal for studying activity features of a large number of neurons during many days of spatial

[1]Spatial Navigation and Memory Unit, National Institute of Neurological Disorders and Stroke, National Institutes of Health, Bethesda, MD 20892, USA. [2]Department of Anesthesiology, Tianjin Medical University General Hospital, Tianjin, China. [3]Integrative Neuroscience Section, National Institute of Mental Health, National Institutes of Health, Bethesda, MD 20892, USA. [4]Office of the Director, National Institute of Mental Health, National Institutes of Health, Bethesda, MD 20892, USA. [5]Present address: Washington University School of Medicine in St. Louis, St. Louis, MO, USA. [6]Present address: Center of Neural Science, New York University, New York, NY, USA. [7]These authors contributed equally: Taylor J. Malone, Nai-Wen Tien, Yan Ma. ✉e-mail: yi.gu@nih.gov

learning. Thus, a reliable and comprehensive measurement of long-term MEC neural dynamics during spatial learning, which could be best achieved using cellular-resolution two-photon imaging[13,14], is needed to fully understand the association between MEC activity features and spatial learning. Furthermore, the necessity of MEC activity in spatial memory remains to be determined by disrupting the map and evaluating behavioral effects after memory is established (Fig. 1a).

In addition, neural dynamics of the MEC during learning are also important for validating theories for the formation of grid cell activity patterns in novel environments. The continuous attractor network (CAN) models predict that grid cell activity is shaped by recurrent connectivity, which is stably maintained within a mature grid network and tightly constrains phase relationships between grid cells (spatial offsets of grid activity fields), regardless of altered network states[15–17]. Although CAN models explain the grid activity pattern per se, they do not answer how grid activity is aligned to a new environment during learning. Other models address this alignment issue by introducing synaptic plasticity of feedforward inputs from cells encoding environmental features onto grid cells, so that grid cell activity can be gradually tethered to new environments and stabilized. Meanwhile, phase relationships between grid cells are preserved by their stable recurrent connectivity[18–27] (Fig. 1b). Testing these models requires measuring neural activity of grid cells and other cells encoding environmental features[28–30] during spatial learning. However, such measurement has not been conducted.

Here, we address the above hypotheses (Fig. 1) using cellular-resolution two-photon imaging[13,14], which enabled reliable tracking of calcium activity in several thousands of MEC neurons across 11 days in mice with different levels of spatial learning performance in virtual reality (VR) environments. Combining the imaging with histology, we discovered that successful spatial learning was associated with

increased c-Fos expression upon novel environment exposure and gradually improved spatial activity consistency. In contrast, unsuccessful spatial learning corresponded to high but unchanged c-Fos expression and an inconsistent cognitive map. These results suggest a synaptic plasticity-based mechanism shaping a consistent MEC map in successful learning. This consistent map is necessary for spatial memory, as optogenetically introducing inconsistent, but not consistent activity, to the MEC specifically impaired mouse behavior once spatial memory was formed. Together, our study reveals a direct link between spatially consistent MEC dynamics and spatial learning and memory. Moreover, we demonstrate that during spatial learning, grid cells increase their spatial tuning consistency while maintaining their phase relationships, suggesting a mechanism involving both synaptic plasticity and stable recurrent connectivity to shape grid cell activity in novel environments.

## Results

### Variable levels of performance in mice during spatial learning

To measure neural dynamics of the MEC during spatial learning, we performed cellular-resolution two-photon calcium imaging in 15 head-fixed Thy1-GCaMP6f transgenic mice (GP5.3)[31,32] while they unidirectionally navigated along one-dimensional (1D) VR tracks, in which visual cues and water rewards were consistently delivered at specific locations (Fig. 2a). GP5.3 mice stably express calcium indicator GCaMP6f in both stellate and pyramidal cell populations in layer 2 of the MEC[33], setting the stage for long-term measurement of calcium dynamics of MEC excitatory neurons. This experimental paradigm allowed us to correlate MEC neural dynamics with mouse behaviors during spatial learning.

Water-restricted naïve GP5.3 mice were first trained to run on a 10-meter 1D VR track (familiar environment, FE) for water rewards (Fig. 2b,

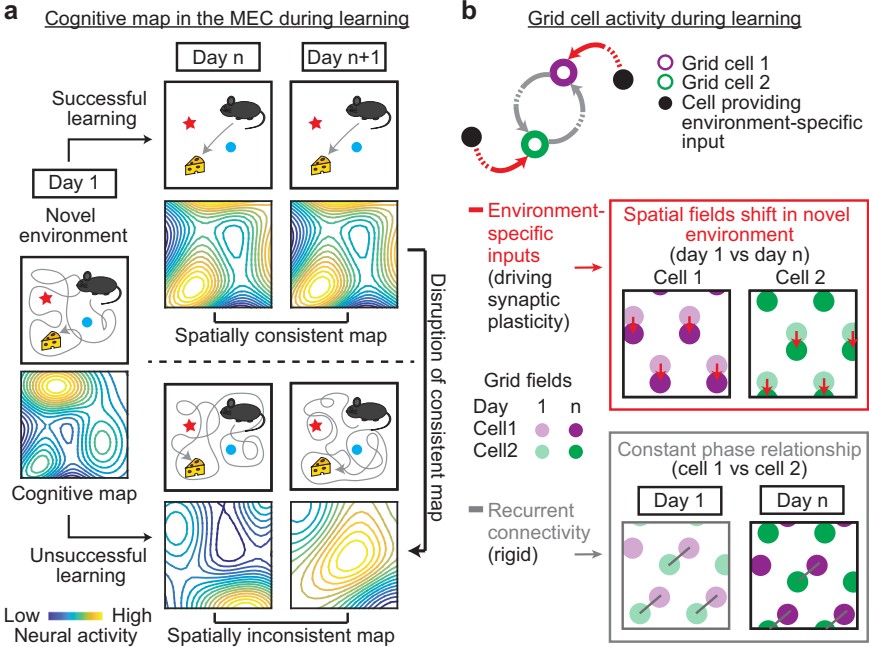

**Fig. 1 | Hypotheses of MEC dynamics during learning. a** Hypothesis of the MEC cognitive map during learning: The map formed after novel environment exposure can change after repeated exposures. After successful learning, the map became spatially consistent across days, whereas unsuccessful learning corresponds to a spatially inconsistent map. Disrupting the consistent map leads to impaired spatial memory, reflected by poor navigation performance. **b** Hypothesis of grid cell activity during learning: (Top) In a novel environment, environmental-specific feedforward inputs (red arrows) drive synaptic plasticity of grid cells and lead to the gradual formation of a consistent grid map. Meanwhile, phase relationships

between grid cells are maintained by rigid recurrent connectivity (gray arrows). (Middle) Spatial fields of individual grid cells shift (downward red arrows) to align with an environment during learning through a synaptic plasticity mechanism driven by environment-specific inputs. (Bottom) Constant phase relationships of grid cells are maintained during learning, as indicated by the same orientation and length of all gray bars connecting corresponding grid fields. In our study, phase relationships of grid cell activity in a two-dimensional (2D) space here were approximated by the correlation of their temporal activity in a one-dimensional (1D) track.

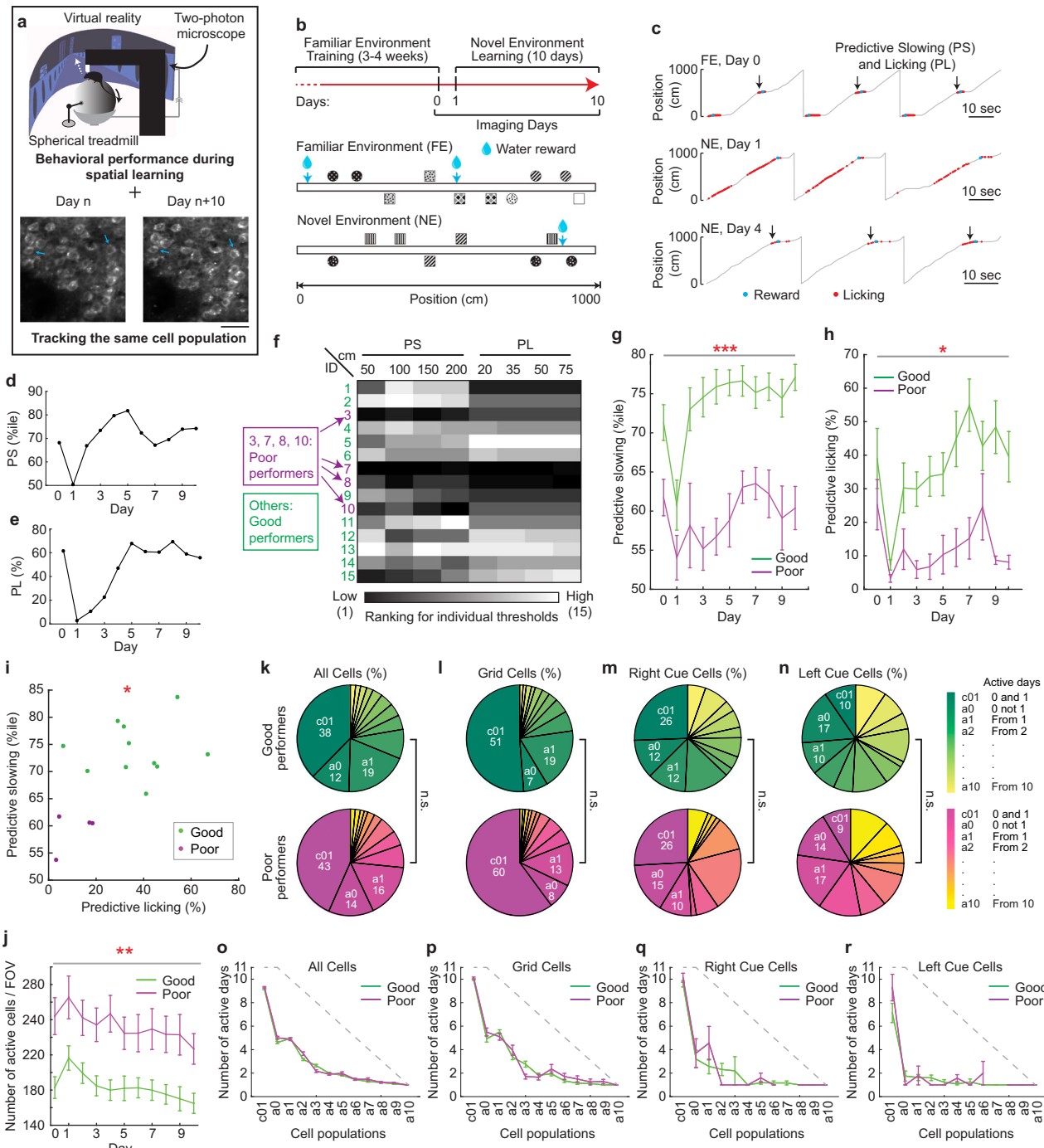

Fig. 2 | **Behavioral performance of mice and dynamics of the active neural ensemble of the MEC during spatial learning in virtual reality. a** Experiment schematic. An example FOV of calcium imaging indicates reliable cell tracking across days (blue arrows). Scale bar: 50 μm. Diagram is adapted from Gu et al.[33,94]. **b** Task schematic. **c** Behavioral examples of mice in FE on day 0 (top) and in NE on day 1 (middle) and day 4 (bottom). Black arrows indicate the locations near rewards where predictive slowing (PS) and licking (PL) occurred. **d**–**e** Example of PS (**d**) and PL (**e**) of one mouse. **f** Ranking of the 15 mice under four distance thresholds for PS and PL. **g**–**h** PS under 150 cm threshold (**g**) and PL under 20 cm threshold (**h**) of good and poor performers, as in all following figures. Significance is expected because the two groups were defined using a combination of the same criteria (PS

and PL) that were used for statistical testing. **i** Correlation between averaged PS and PL over 10 days in the NE. **j** Number of active cells per FOV. **k**–**n** Fractions of all cells (**k**), grid cells (**l**), right cue cells (**m**), and left cue cells (**n**) active on different days. **o**–**r** Number of days the individual cell populations in (**k**–**n**) stayed active during the 11 days (**o** – all cells; **p** – grid cells; **q** – right cue cells; **r** – left cue cells). Gray dashed lines represent the maximum number of days a cell could be active based on the first day identified. *$p \leq 0.05$, **$p \leq 0.01$, ***$p \leq 0.001$. Error bars represent mean ± sem. Horizontal gray lines indicate $p$ values for the group difference. See Supplementary Data 1 for exact $n$ and detailed statistical information. Source data are provided as a Source Data file. See also Figs. S1 and S2.

S1a). After 3–4 weeks of training, mice generally developed predictive slowing (PS) and licking (PL) prior to rewards, indicating the antici-pation of learned reward locations (Fig. 2c)[34–37]. Either high PS or PL was taken as an indicator of environmental familiarity. Once mice

exhibited stable PS and PL (see Methods), they were exposed to a novel environment (NE) with a reward and new cues arranged at different locations compared to the FE (Figs. 2b, S1b), and their behavior in the NE was monitored for ten consecutive days. Expectedly, PS and PL

generally decreased on day one in the NE and improved with increased experience (Fig. 2c–e), indicating learning of the NE.

We ranked learning performance of individual mice in the NE based on their PS and PL calculated using multiple distance thresholds (Fig. S1c; see Methods). Four mice (3, 7, 8, and 10) were consistently ranked in the lower half of the cohort in all conditions (Fig. 2f) and, therefore, were categorized as relatively poor performers. The others were categorized as relatively good performers. The good performers showed higher PS and PL than the poor performers during learning (Fig. 2g, h), and averaged PS and PL of individual mice in the NE were positively correlated (Fig. 2i). The good performers generally achieved stable performance after 6.3 days in the NE (an average of 4.5 days for PS and 8 days for PL, Fig. S1d, e).

Further analyses indicate that PS and PL reflected a mouse's spatial learning ability, rather than other factors, such as attention, experience, environmental features, age, or sex (Fig. S1f–o). The good and poor performers also learned equally well in a visual discrimination task to identify the correspondence between cue patterns used in the NE track and reward delivery (Fig. S1p–s), indicating no difference in their ability to distinguish visual patterns and to learn associations between patterns and rewards. Therefore, the good and poor performer groups, which represented successful and unsuccessful spatial learning, respectively, were used in subsequent analyses.

## Properties of the active neural ensemble of the MEC during spatial learning

We analyzed MEC calcium dynamics in the 15 mice underlying their different spatial learning performances. Active cells in individual imaging fields of view (FOVs) were longitudinally tracked for 11 days (one day in the FE (day 0) and ten days in the NE (days 1–10); Fig. 2b). Overall, the poor performers showed larger numbers of active cells per FOV than the good performers (Fig. 2j). Given that the two performer groups had comparable numbers of visibly identifiable cells, including pyramidal and stellate cells, this difference reflected a higher percentage of active cells in both cell populations in the poor performers (Fig. S2a–d). Both performer groups exhibited a significant expansion of the active cell population on day one in the NE, which slowly shrank during learning (Fig. 2j, S2e).

We next asked whether the decreased number of active cells during learning reflected different neural populations becoming active on each day, or the refinement into a smaller cell population. We classified active cells into 12 categories: those commonly active on days 0 (in the FE) and 1 (in the NE) (c01), active on day 0 but not day 1 (a0), and newly active on day 1, day 2, …, and day 10 (a1, a2, …, and a10). Of these categories, the c01 cells were the largest population (Fig. 2k) and were most persistently active in both performer groups (Fig. 2o). The fraction of newly active cells gradually decreased during learning (Fig. 2k). To eliminate the possibility that the features of c01 cells were due to the specific selection of cells active on two adjacent days, we investigated other cell categories, the first two adjacent active days of which were days 1 and 2 (c12), 2 and 3 (c23), … and days 9 and 10 (c910). Compared to these cells, c01 cells were still the largest population (Fig. S2f) with the most persistent activity across days (Fig. S2g). These dynamics together indicate that the c01 cells were stabilized during learning, whereas other cells were gradually eliminated.

We next focused on grid and landmark cells because both cell types are potentially involved in spatial learning[7,9–11,26,27]. Cue cells are MEC landmark cells in VR and are specifically active around left or right visual landmarks (left or right cue cells, respectively)[29,38]. Since the number of active days varied for individual neurons (Fig. 2k, o), we defined true grid and cue cells (left or right) as those classified as such on more than half of their active days based on previous classification criteria[29,33,39]. Overall, both performer groups had comparable percentages of grid cells, right cue cells, and left cue cells (Fig. S2h). The

c01 cells contained the largest fractions of grid and right cue cells, and around 10% of left cue cells (Fig. 2l–n), all of which were the most stably active populations across days (Fig. 2p–r).

Finally, we examined the anatomical clustering of the c01 cells relative to "other cells" (a0 through a10) (Fig. S2i), as previous work showed that functionally relevant MEC cells, such as grid cells, are anatomically clustered[13,33,40]. Although c01 cells in both performer groups showed clustering relative to "other cells", in the good performers they were clustered both locally and globally, whereas in the poor performers they were only clustered locally (Fig. S2j–n).

Overall, regardless of learning performance, c01 cells were the largest and most persistently active population during learning, whereas other cells were gradually eliminated, suggesting the refinement of the MEC spatial map. Therefore, the c01 population was the best candidate to be linked to the spatial memory.

## Higher inter-day activity consistency during learning in good performers

We further focused on the subset of the c01 cells that were persistently active for all 11 days (persistent cells) to investigate the relationship between MEC dynamics and learning performance of the mice. Around 80% of the persistent cells were stellate cells (Fig. S3a), which constituted less than 60% of GCaMP6f+ excitatory cells (Fig. S3b). This enrichment of stellate cells in the persistent cell population supports their essential role in spatial learning[41].

We first investigated the persistent cells in their overall calcium response ($\Delta F/F$), which was represented by statistically significant calcium transients. Compared to the good performers, the poor performers showed lower $\Delta F/F$ (Fig. 3a), resulting from the low frequency of their significant transients, despite the higher amplitudes and longer durations of the transients (Fig. S3c–e). $\Delta F/F$ of both performer groups showed a comparable increase on day one in the NE and gradually decayed during learning (Fig. 3a, S3f). This observation and the rapid expansion of the active cell population on day 1 (Fig. 2j) indicate that the MEC network of both performer groups immediately responded to NE exposure.

Consistent with previous reports[42], NE exposure induced global activity remapping in the MEC in both performer groups, as shown by the dissimilarity of their population activity matrices on days 0 and 1 (Fig. 3b, c, the first two columns), and the nearly-zero correlations between population activity matrices and between spatially binned calcium responses of individual cells on days 0 and 1 (Fig. 3d, e). These low correlations were uncorrelated with PS and PL of individual mice (Fig. S3g, h) and occurred in both performer groups on the very first run in the NE (Fig. S3i, j). These results indicate that the good and poor performers exhibited similar levels of global remapping upon NE exposure.

We next evaluated inter-day consistency by calculating the correlation of population activity matrices on adjacent days. The correlation for the good performers gradually increased in the NE during learning, whereas for the poor performers it stayed low and did not improve (Fig. 3b–d). The same difference existed when correlating spatially binned calcium responses of individual cells (Fig. 3e). These correlations were also positively correlated with averaged PS and PL on adjacent days of individual mice, further supporting the association between the inter-day activity consistency and learning performance (Fig. S3k, l).

We next asked what activity pattern the MEC map had acquired during learning by focusing on the percentage of the track covered by spatial fields (spatial field coverage), which consist of adjacent spatial bins with significant calcium responses. Higher spatial field coverage generally reflects more spatially frequent and consistent neural response (Fig. S3m). After an initial decrease upon entering the NE, the field coverage in the good performers steadily increased and stabilized during learning (Fig. 3f). Despite having comparable anatomical

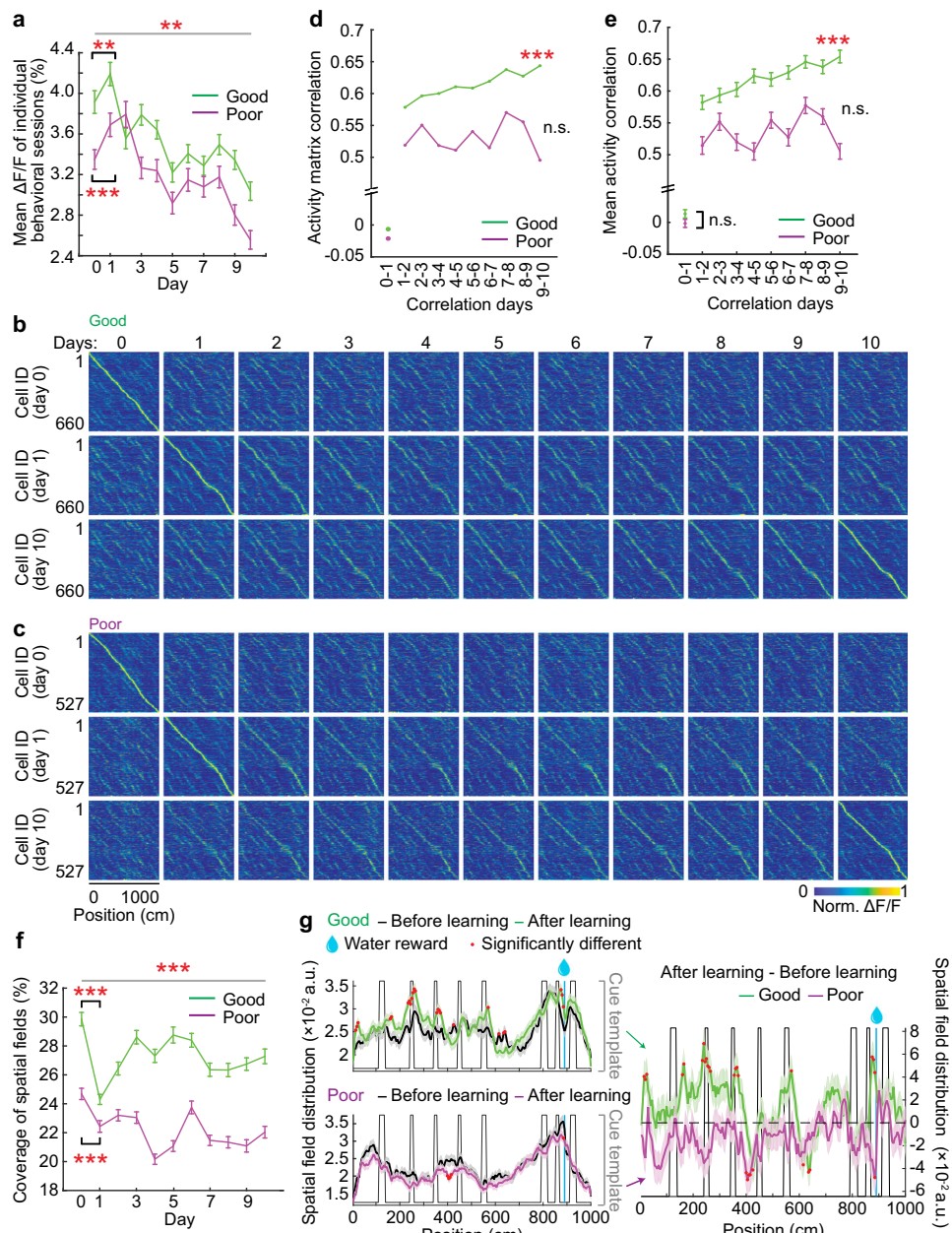

**Fig. 3 | Inter-day activity consistency of MEC neurons during learning. a** Mean calcium response (mean ΔF/F) of MEC neurons during learning. **b–c** Activity matrix of good (**b**) and poor (**c**) performers. The normalized mean ΔF/F of individual cells were sorted based on the peak locations on days 0 (top row), 1 (middle row), and 10 (bottom row). **d–e** Correlation of activity matrices (**d**) or of mean activity (ΔF/F) as a function of track position for individual cells (**e**) on adjacent days. * indicates the significant positive or negative correlation from days 1–2 to days 9–10. **f** Percentage

of track covered by spatial fields. **g** Left: Spatial field distribution before (days 1 and 2, black) and after learning (days 7–10, colored). Right: Change in spatial field distribution after learning (difference between the curves on the corresponding left panel). *$p \leq 0.05$, **$p \leq 0.01$, ***$p \leq 0.001$. Error bars and shading represent mean ± sem. Horizontal gray lines indicate $p$ values for the group difference. See Supplementary Data 1 for exact $n$ and detailed statistical information. Source data are provided as a Source Data file. See also Fig. S3.

locations along the ventral-dorsal axis of the MEC (Fig. S3n), which could affect spatial tuning features[43,44], spatial field coverage in the poor performers was overall lower and did not increase during learning (Fig. 3f). Spatial field coverage was also positively correlated with daily PS and PL on an individual-mouse level (Fig. S3o). The higher spatial field coverage in the good performers during learning resulted from increased spatial fields around several cues and before the reward (Fig. 3g), indicating enhanced representation of goal locations[9,10] and salient landmarks by the MEC map during learning. This change was not observed in the poor performers.

These results demonstrate that during successful learning, the MEC spatial map became consistent across days, and the map

represented salient environmental features, despite overall decreased amplitude of the neural response. In contrast, unsuccessful learning was coupled with a less reliable spatial map on a day-by-day basis.

## Higher intra-day activity consistency and decoding ability in good performers

We further investigated intra-day activity consistency of individual cells by calculating activity correlation on a run-by-run (RBR) basis within a single behavioral session of a day. The good performers showed decreased RBR correlation on day one in the NE, and the correlation increased and then stabilized around day seven (Fig. 4a, c).

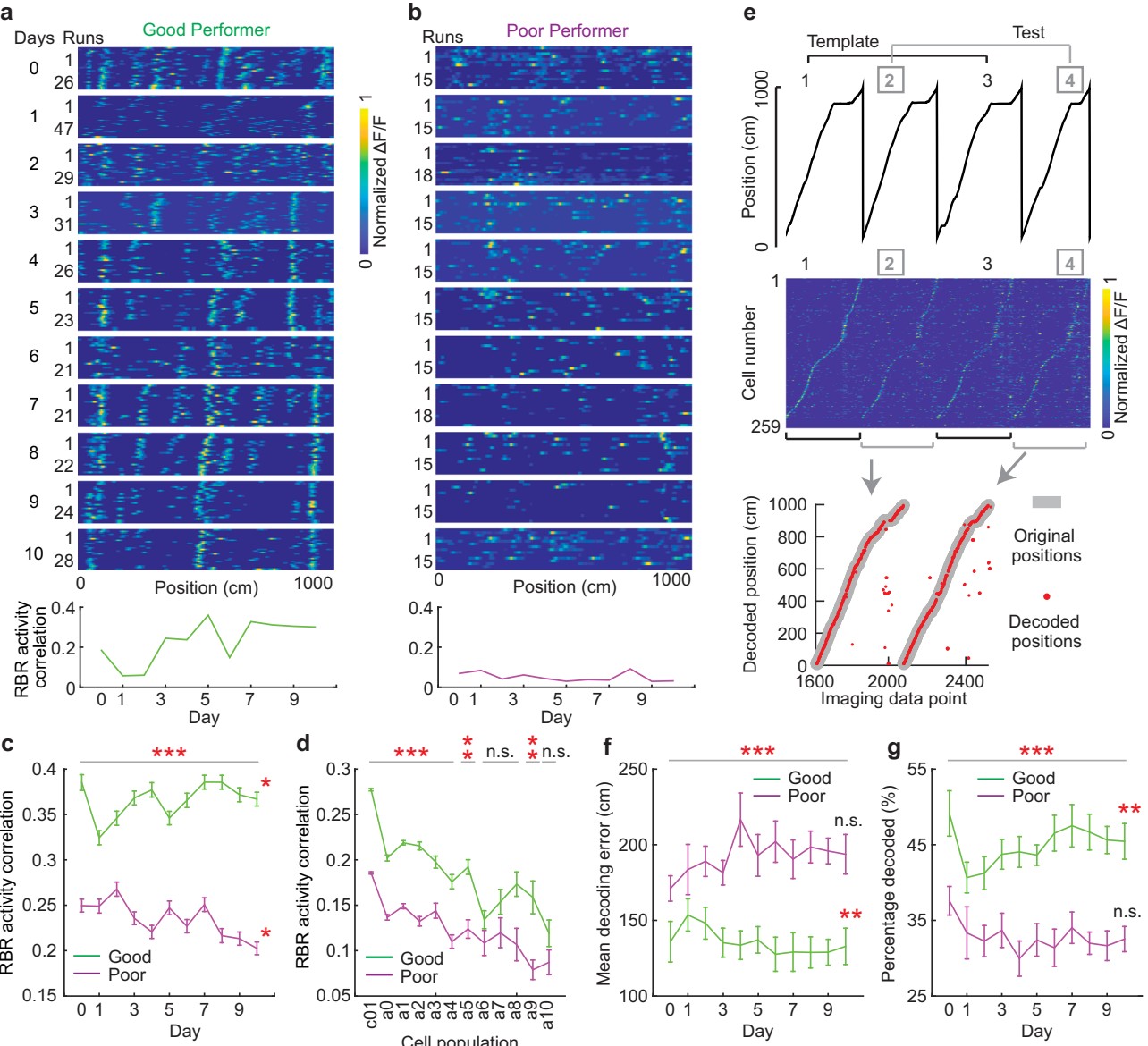

**Fig. 4 | Intra-day activity consistency of MEC neurons during learning.**
**a–b** Panels 1 to 11 (from top to bottom): Example RBR calcium response of a good (**a**) or poor (**b**) performer cell from days 0 to 10. Bottom Panel: RBR activity correlation of example cell. **c** RBR activity correlation of the cells active for 11 days. **d** RBR correlation of cells in different activity categories, as identified in Fig. 2k. The correlation for each cell was the mean RBR activity correlation of all its active days. * indicates the statistical significance for cells in different categories. **e** Decoding of track positions based on population activity. Top: Four example runs. The odd and even runs were used as template (black) and test runs (gray), respectively. Middle: population activity in the four runs, sorted based on the peak activity locations in run 1. Bottom: a comparison of the original (gray) and decoded positions (red) in test runs. **f–g** Mean decoding error (**f**) or percentage of correctly decoded track positions (**g**) using 50 randomly selected cells. For each FOV, two random selections of 50 cells were made and the results were averaged. *$p \leq 0.05$, **$p \leq 0.01$, ***$p \leq 0.001$. Error bars represent mean ± sem. Horizontal gray lines indicate $p$ values for the group difference, except as noted. * to the right of line graph indicates significant positive or negative correlation from days 1 to 10. See Supplementary Data 1 for exact $n$ and detailed statistical information. Source data are provided as a Source Data file. See also Fig. S4.

In contrast, the cells of the poor performers had lower RBR correlations that further decreased during subsequent days (Fig. 4b, c). RBR correlation was positively correlated with daily PS and PL of individual mice (Fig. S4a). Cells becoming active on different days or on different adjacent pairs of days, as categorized in Fig. 2k and Fig. S2f, generally had high RBR activity correlation in the good performers. Among these subpopulations, c01 cells had the highest RBR activity correlation (Figs. 4d, S4b), indicating that c01 cells not only were the largest and most persistently active cell population, but also had the highest spatial activity consistency.

We further examined the ability of the MEC population activity to decode VR track locations using a population-vector-based method[45]

(Fig. 4e). Because the performance of the decoder was sensitive to the number of cells used in the analysis (Fig. S4c)[45], we evaluated decoding accuracy by assessing 50 simultaneously imaged cells randomly chosen from each FOV, and calculated decoding error and the percentage of correctly decoded track positions. The good performers showed smaller decoding errors and a higher percentage of decoded track positions compared to the poor performers (Fig. 4f, g). The decoding accuracy of the good performers improved and stabilized during learning, while the poor performers showed poor decoding performance without improvement. Decoding accuracy was correlated with daily PS and PL of individual mice (Fig. S4d, e). When evaluating the decoding performance using all active cells, the poor performers still

exhibited relatively lower decoding accuracy despite their higher numbers of active cells (Figs. 2j, S4f, S4g).

These results indicate that during successful but not unsuccessful learning, the MEC spatial map showed high and improved intra-day spatial consistency and accurate spatial decoding. Higher running speed has been reported to increase precision of MEC spatial encoding[46], but running speed was not different in the two performer groups (Fig. S4h) and did not contribute to spatial encoding differences (Fig. S4i–t).

### Different spatial dynamics of grid and cue cells during spatial learning

Next, we focused on activity dynamics of grid and cue cells among the cells persistently active for 11 days. The percentages of grid and cue cells were comparable in the two performer groups (Fig. S5a–d). The percentages of left cue cells were lower than right cue cells (Fig. S5d), consistent with the dominant representation of right cues in the left MEC[29]. Therefore, left and right cue cells were combined for following analyses.

After global activity remapping on day one for both performance groups, grid cells of the good performers, but not the poor performers, showed higher and improved consistency in their inter- and intra-day activity (Figs. 5a, c, e and S5e, h, S6a, c), reflecting a gradual stabilization of the grid map. The inter- and intra-day activity consistency of grid cells was also positively correlated with PS and PL of individual mice (Figs. S5k, l, S6e, f). In agreement with prior research[7], grid scale increased upon entering the NE, and gradually decreased over the subsequent days specifically in the good performers (Fig. S5m). The grid scale expansion on day one could reflect the decreased number of grid fields (Fig. S5n) that led to larger spacing between adjacent fields. In contrast, cue cells in both performer groups constantly represented landmark patterns in both the FE and NE. Their activity had stable inter- and intra-day consistency, which was higher than that in grid cells (Figs. 5b, d, f, S5f, g, i, j, S6b, d). Cue cells in the poor performers, however, showed less consistent intra-day activity than the good performers (Fig. 5f). Given that single cue cells typically responded to individual cues with a constant spatial shift but with different amplitudes (Fig. S6g)[29], we further analyzed these features on an RBR basis. While cue cell responses showed small spatial shift deviations across runs in both performer groups (Fig. S6h), they had greater amplitude variations to individual cues in the poor performers (Fig. S6i, j).

Overall, the grid map gradually stabilized during successful learning, whereas cue cells reliably represented landmarks across environments and throughout learning. These dynamics are in line with the hypothesis that landmark signals guide the formation of a consistent grid map in a novel environment through a synaptic plasticity-based mechanism (Fig. 1b). During unsuccessful learning, cue cells were unable to maintain reliable response amplitudes to individual cues, which could potentially affect grid map consistency.

### Constant phase relationships of grid cells during learning

We further asked that during the gradual change in spatial activity consistency of grid cells in the NE, whether phase relationships of spatial activity of co-modular grid cells, which are grid cells with similar scales[44], remained stable (Fig. 1b). We examined phase relationships of simultaneously imaged co-modular grid cells, which were determined based on similar spacings and widths of their spatial fields (see Methods). A previous study demonstrated that small and large phase differences of grid cell pairs correspond to high and low correlations of their temporal activity, respectively[47]. Therefore, we used pairwise correlations (Pairwise corr.) of 1D temporal calcium activity of grid cells to approximate their 2D phase relationships (Fig. 5g). We first calculated Pairwise corr. of all simultaneously imaged co-modular grid pairs on

individual days. We then compared their Pairwise corr. on adjacent days to evaluate whether their Pairwise corr. patterns, i.e., phase relationships, were preserved across days (Fig. 5h). Indeed, Pairwise corr. of the good performers on adjacent days showed high and stable correlations (Day corr.) (Fig. 5i, k), indicating stable phase relationships of co-modular grid cells on adjacent days, regardless of the environmental switch and subsequent learning. In addition, Day corr. between grid cells in different modules (grid cells with different scales) were lower than those within the same module (Fig. S7a, c), as expected if different grid modules belong to different attractor networks and are less connected[44,47,48]. In contrast, grid phase relationships in the poor performers were weakly preserved within the same module (Fig. 5j, k) and the difference in phase relationships between the same and different modules was diminished (Fig. S7b, d).

We further asked whether phase relationships of grid cells were stable enough to maintain their topographical organization on the MEC cell sheet throughout learning, as the consistency between anatomical arrangement of grid cells and their connectivity would facilitate efficient functioning of the circuit[33]. In layer 2 of the MEC, grid cells are anatomically arranged according to their phases: within an anatomical distance of 120 μm, phase differences between co-modular grid cells increase with distance. This pattern repeats on the MEC cell sheet as a 2D phase lattice[33]. Similar to above, this topographical organization of grid phases can be revealed by evaluating pairwise activity correlation of grid cells as a function of their anatomical distance, i.e., grid cells in close proximity (~0–50 μm) and at large distances (~200–250 μm) have highly correlated activity, whereas those at intermediate distances (~100–150 μm) have poorly correlated activity[13] (Fig. 5l). Therefore, we examined whether the relationship between activity correlation (the same Pairwise corr. calculated above) and anatomical distance between grid cells was maintained during learning. We calculated mean Pairwise corr. of cell pairs within three distance ranges: zone 1: 0–50 μm, zone 2: 92–132 μm, and zone 3: 193–245 μm. In the good performers, the Pairwise corr. in zones 1 and 3 were higher than those in zone 2, and this pattern was observed before (days 1–2) and after (days 7–10) learning (Fig. 5m, n) and on individual days (Fig. S7e, f). In addition, there was an overall increase in Pairwise corr. between grid cells during learning (Fig. 5n, 5o, S7g). To eliminate the effect of this increase, we further calculated "Adjusted Pairwise corr." by subtracting the mean Pairwise corr. of grid cell pairs at all distances from the Pairwise corr. in the three zones. Adjusted Pairwise corr. in each of the three zones was comparable before and after learning (Fig. 5p), as well as on individual days (Fig. S7h). These observations suggest the grid cell network receives feedforward input that is strengthened during learning, leading to a network-wide increase in activity correlations of grid cell pairs. Meanwhile, rigid recurrent connectivity between grid cells maintains their phase relationships throughout learning. In contrast, the poor performers did not show the characteristic correlations in the three zones (Figs. 5q, r, t, S7i, j, l), and only exhibited a slight increase in the overall Pairwise corr. between grid cells during learning (Figs. 5r, s, S7k).

Overall, during the learning of the good performers, although grid cells exhibited increased spatial tuning consistency and temporal activity correlation, their phase relationships remained constant, strongly supporting stable recurrent connectivity between grid cells throughout learning. However, the grid network coherency was disrupted in the poor performers.

### c-Fos expression in the good and poor performers

The stabilization of a consistent MEC map during learning suggests the involvement of synaptic plasticity that is induced upon novel environment exposure and shapes the new spatial map. To test this

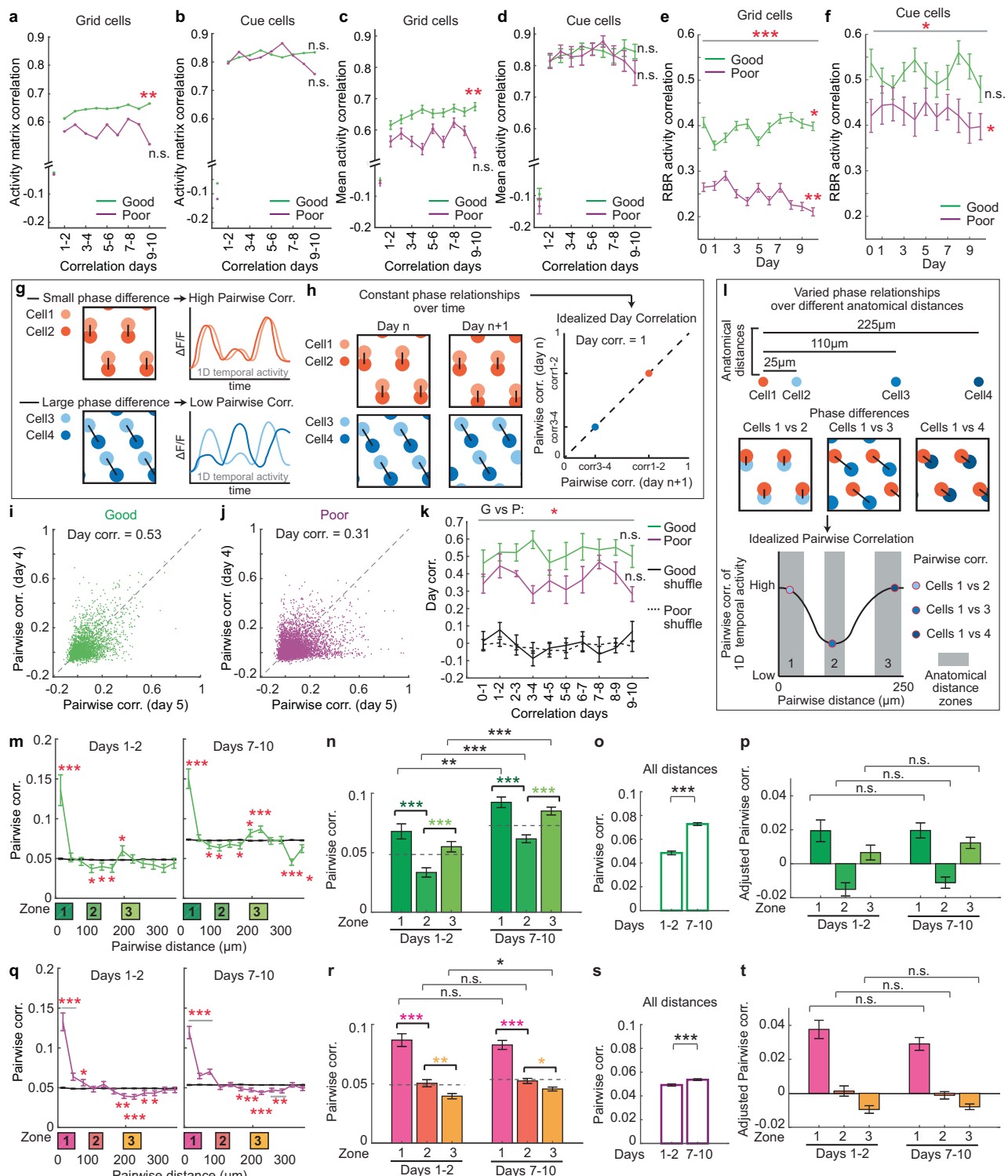

hypothesis, we analyzed expression of an immediate early gene, c-Fos, which indicates neuronal activation and is also linked to neurons undergoing learning-induced synaptic plasticity for memory encoding[49–51]. We histologically examined c-Fos protein expression in MEC layer 2, the same region imaged above, of 20 mice (histology mice) after a 1-h exposure to FEs or NEs (Fig. 6a). The 11 good and 9 poor performers among histology mice were grouped based on their PS and PL in FEs using the thresholds developed from the behaviors of imaging mice in their last six days in the NE (see Methods). Histology mice showed comparable learning performance to the corresponding

groups of imaging mice (Fig. 6b, c). The good performers exhibited low c-Fos levels in FEs and high levels in NEs (Fig. 6d, e), consistent with previous observations that c-Fos expression in the MEC increased during early training of learning tasks and decreased with extended training[52,53]. In contrast, the poor performers had a high c-Fos level in both FEs and NEs. Most c-Fos+ cells were reelin+ stellate cells, in contrast to the low fractions of calbindin+ pyramidal cells and GAD67+ interneurons (Fig. 6f, g), consistent with stellate cells being the most abundant cell type in the persistent cell population during learning (Fig. S3a) and crucial for spatial learning[41].

**Fig. 5 | Grid and cue cell activity during learning. a–b** Activity matrix correlations on adjacent days for grid (**a**) and cue (**b**) cells. **c–d** Correlation of mean ΔF/F as a function of track position on adjacent days for individual grid (**c**) and cue (**d**) cells. **e–f** RBR activity correlation of grid (**e**) and cue (**f**) cells. **g** Phase relationships of grid cells as represented by pairwise correlations (Pairwise corr.) of their temporal calcium activity. Small and large phase differences of grid cell pairs (left) correspond to high and low correlations of their temporal activity (right), respectively. **h** If pairwise phase relationships between grid cell pairs are preserved on adjacent days (left), the Pairwise corr. of these pairs on adjacent days will be perfectly correlated (Day correlation/Day corr. = 1; right). **i–j** Pairwise corr. of grid activity (ΔF/F) on days 4 and 5 in the good (**i**) and poor (**j**) performers. **k** Day corr. between adjacent days for the good performers (G), poor performers (P), and their shuffles (Good or Poor). Error bars are generated by data from individual FOVs. **l** The phase relationship of grid cell pairs varies with respect to their anatomical distance. Cells in close proximity (zone 1: 0–50 μm) and farther distances (zone 3: 193–245 μm) have small phase differences (top) and therefore high Pairwise corr. (bottom). Grid cells at intermediate distances (zone 2: 92–132 μm) have large phase differences

(top) and therefore low Pairwise corr. (bottom). Gray regions represent the above anatomical distance zones 1–3 used for the following analysis. **g, h, l** For clarity, we show 2D grid cell relationships here. In this study, we use the correlation of 1D temporal activity to approximate 2D phase relationships. **m, q** Pairwise corr. as a function of pairwise distance for the good (**m**) and poor (**q**) performers before (days 1–2) and after (days 7–10) learning. * indicates the significant difference between real data and shuffles. **n, r** Pairwise corr. in the three zones before (days 1–2) and after (days 7–10) learning in good (**n**) and poor (**r**) performers. **o** and **s** Comparison of the pairwise correlations of grid cell pairs in the good (**o**) and poor (**s**) performers at all distances before and after learning. **p, t** Comparison of Adjusted Pairwise corr. in the three zones before and after learning in the good (**p**) and poor (**t**) performers. *$p \leq 0.05$, **$p \leq 0.01$, ***$p \leq 0.001$. Error bars represent mean ± sem. Horizontal gray lines indicate $p$ values for the group difference. * to the right of line graph indicates significant positive or negative correlation from day 1–2 to day 9–10 or day 1 to day 10, as appropriate. See Supplementary Data 1 for exact $n$ and detailed statistical information. Source data are provided as a Source Data file. See also Figs. S5–S7.

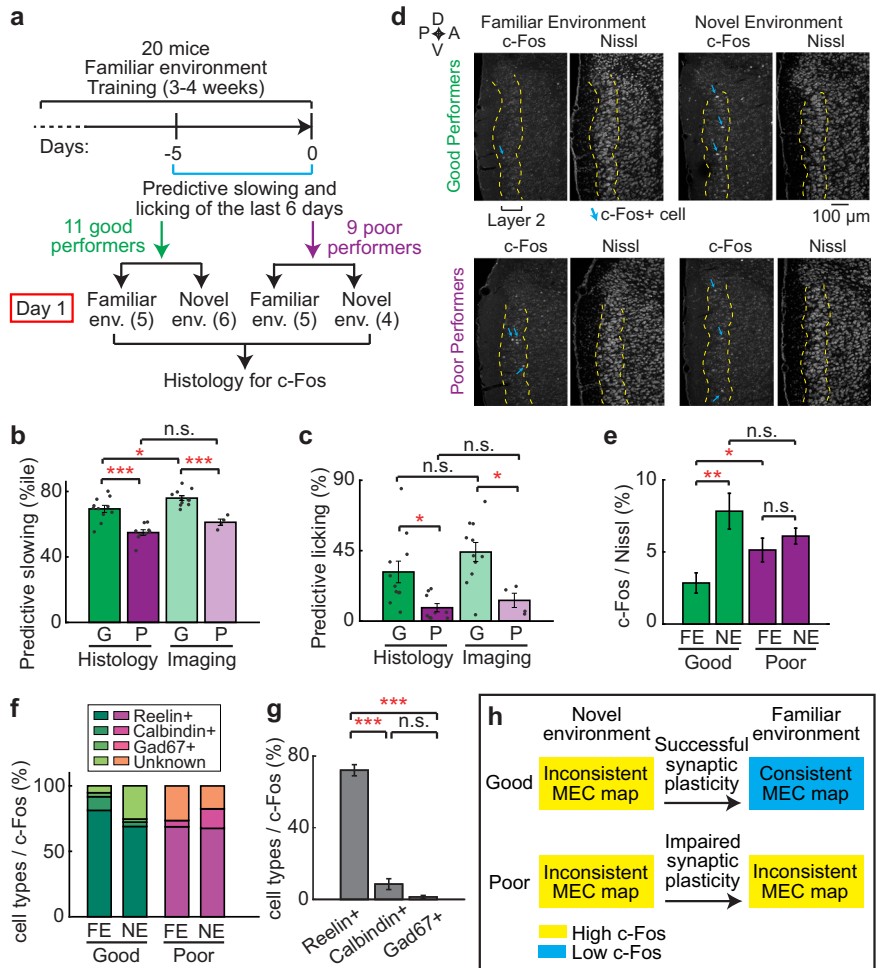

**Fig. 6 | Good performers have increased c-Fos expression in the MEC upon novelty exposure. a** Schematic for histology experiment. The numbers under Day 1 represent the number of mice in familiar and novel environments (env.). **b–c** Comparison of PS (**b**) and PL (**c**) between good (G) and poor (P) performers of imaging and histology mice. **d** Histological staining of c-Fos in the MEC for good and poor performers in familiar and novel environments. Scale bar: 100 μm. D: dorsal; V: ventral; A: anterior; P: posterior. **e** The percentage of cells in layer 2 of the

MEC that are c-Fos + . FE and NE are familiar and novel environments, respectively. **f** Percentages of c-Fos+ cells that are Reelin + , Calbindin + , Gad67+ or uncharacterized cells. **g** The percentage of Reelin+ cells is significantly larger than all the other cell types. **h** Interpretation of the c-Fos result. *$p \leq 0.05$, **$p \leq 0.01$, ***$p \leq 0.001$. Error bars represent mean ± sem. See Supplementary Data 1 for exact $n$ and detailed statistical information. Source data are provided as a Source Data file.

The increased c-Fos expression when the good performers were switched from FEs to NEs reflected an MEC cell population strongly responding to novelty stimuli. This observation, together with changed neural activity in the NE (Figs. 2–4), suggests the

induction of synaptic plasticity to shape a consistent MEC map in the NE. In contrast, in the poor performers, c-Fos level was high in FEs and did not further increase in NEs. Such high basal activity in FEs indicated by c-Fos could lead to dysregulated synaptic

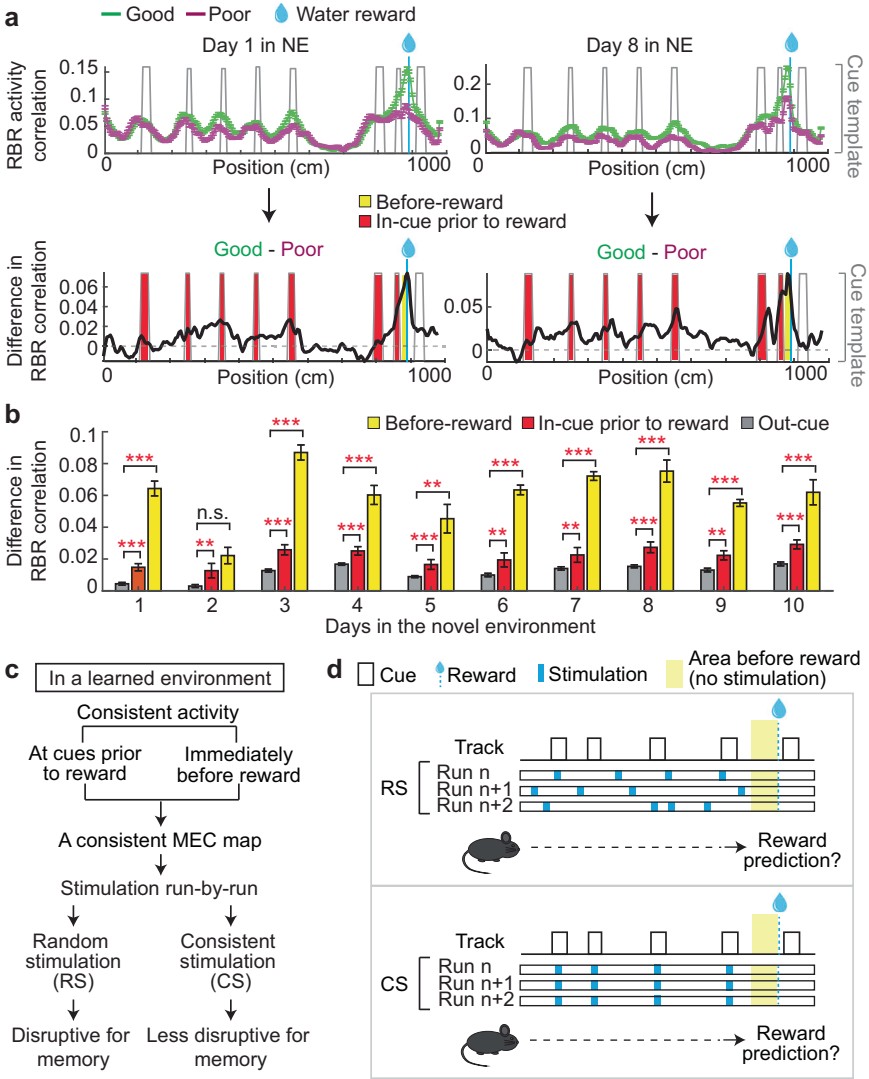

**Fig. 7 | Consistent MEC activity along the track. a** Top: Comparison of RBR spatial activity consistency along the track between the good and poor performers on days 1 (left) and 8 (right). Bottom: The difference between the RBR correlations of the good and poor performers. NE Novel environment, FE Familiar environment. **b** RBR activity correlation difference between the good and poor performers. **c** Hypothetical results of modulating MEC map consistency. **d** Strategy for optogenetics experiments. *$p \leq 0.05$, **$p \leq 0.01$, ***$p \leq 0.001$. Error bars represent mean ± sem of all cells in the good and poor performers. See Supplementary Data 1 for exact $n$ and detailed statistical information. Source data are provided as a Source Data file. See also Fig. S8.

plasticity[54,55], and therefore, an inconsistent MEC map during learning (Fig. 6h).

## Consistent MEC activity is necessary for spatial memory of a learned environment

We next examined whether spatial memory of a learned environment requires a spatially consistent MEC map. To better understand RBR activity consistency along the track, we calculated the consistency as a function of track location in the 10 m NE. In both performer groups, higher consistency was observed at cues preceding the reward and immediately before the reward (Fig. 7a), and these areas also had the largest difference in activity consistency between the good and poor performers throughout learning, compared to the areas outside cues and away from the reward (out-cue areas) (Fig. 7b). This suggests that highly consistent activity patterns at cues preceding the reward and immediately before the reward are important components of the cognitive map that supports spatial memory. Therefore, we hypothesized that disrupting the map consistency on an RBR basis by introducing external

stimulation of MEC neurons at random track locations (random stimulation, RS) should impair spatial memory. In contrast, imposed stimulation of MEC neurons at consistent locations (consistent stimulation, CS), specifically, at the cues prior to the reward or immediately before the reward, would mimic the MEC map consistency and, therefore, be less disruptive to spatial memory (Fig. 7c). Different effects of RS and CS would strongly support the necessity of a consistent MEC map for spatial memory. To test these hypotheses, we optogenetically stimulated the MEC with different patterns (RS or CS) on an RBR basis while mice navigated a previously learned environment. We introduced RS in the areas away from the reward and compared its behavioral effect with that of CS, which was imposed at the cues prior to the reward (Fig. 7d). Although the biggest difference in RBR consistency between the performers was observed in the area immediately before the reward (Fig. 7b), we left the area unmodulated so that after experiencing the modified map prior to the reward, mice could voluntarily initiate reward-predictive behaviors without direct disruption. Additionally, the potentially increased neural response upon

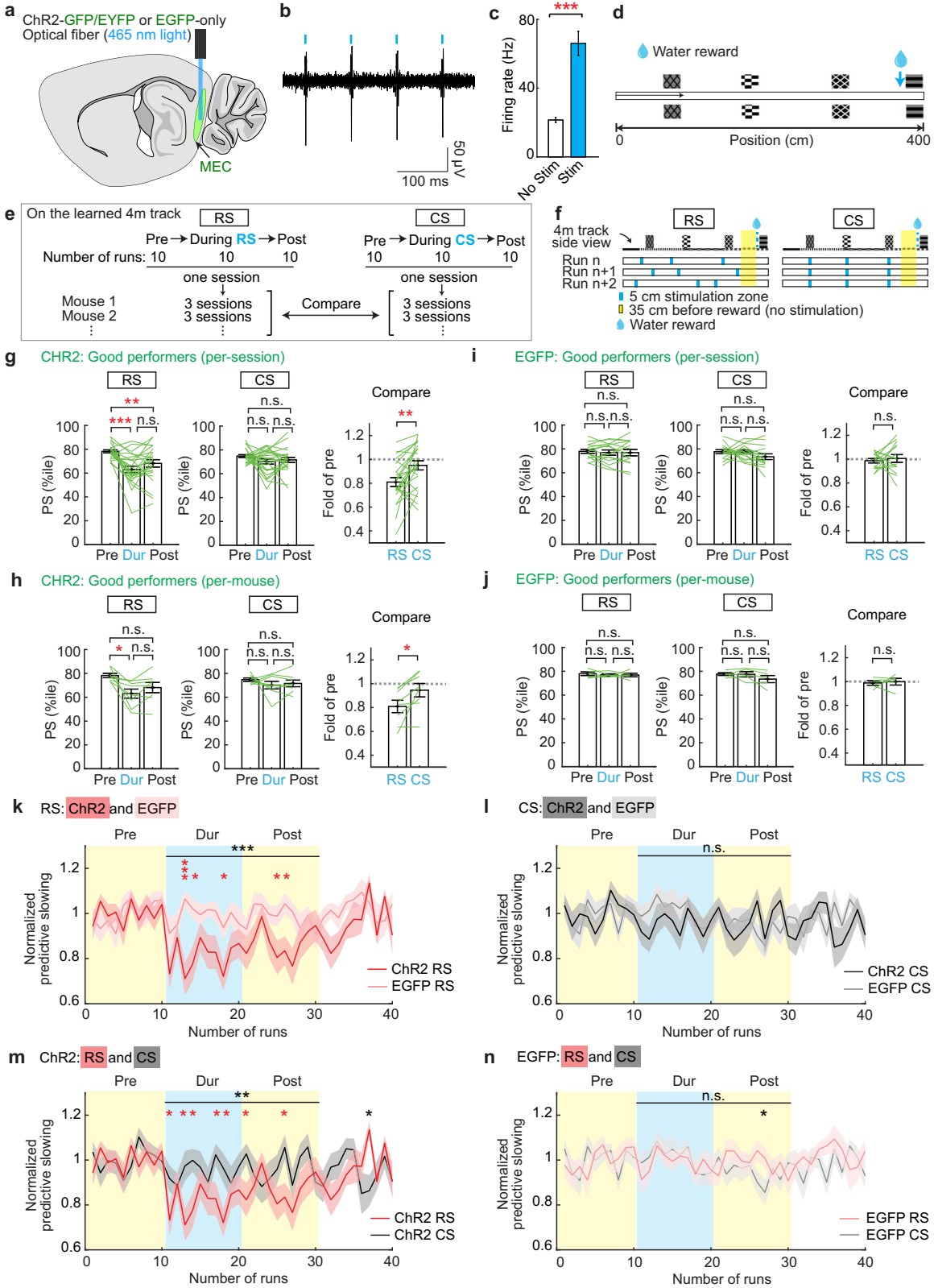

optogenetic stimulation could better mimic the higher calcium response amplitude at cues, but not the lower response immediately before the reward, in comparison to that in out-cue areas (Fig. S8).

Optogenetic activation was achieved by virally expressing channelrhodopsin-2 variant H134R[56] (ChR2-GFP or ChR2-EYFP) in MEC layer 2, and an EGFP-only virus was used as an illumination control

(ChR2 or EGFP mice, respectively, Figs. 8a, S9a). This approach predominantly labeled reelin+ stellate cells (Fig. S9b, c), consistent with the enrichment of stellate cells in the persistently active cell population during learning (Fig. S3a, b). Reliable light-induced action potentials in the MEC of ChR2 mice were confirmed using in vivo multi-unit recordings (Fig. 8b, c). To obtain sufficient runs for behavioral comparison before, during and after stimulation, mice were trained on a

**Fig. 8 | Consistent MEC activity is necessary for spatial memory. a** Schematic displaying the location of the optical fiber implant and blue LED light delivery to activate ChR2-GFP/EYFP or EGFP expressing neurons in the superficial layers of the MEC. Sagittal section is adapted from ref. 95. **b** In vivo extracellular recording of light-evoked spikes from an anesthetized ChR2-GFP mouse. **c** Average change in the number of spikes during stimulation (Sim) versus no stimulation (No Stim) trials across one full recording session from an anesthetized ChR2-GFP mouse. **d** Top-down view of the 4 m track. **e** Experiment design. **f** Schematics showing the stimulation of three randomly located 5 cm zones (RS) and consistent stimulation for 5 cm at each cue preceding the reward (CS) on an RBR basis. **g–j** Predictive slowing (PS) of ChR2 (**g** and **h**, 8 mice) and EGFP (**i** and **j**, 5 mice) good performers pre-, during- (Dur), and post- the RS (left) and CS (middle) in **f**. Data was calculated per session (**g** and **i**, 3 sessions per mouse) and per mouse (**h** and **j**). Mean PS of 10

runs in the three periods are plotted. Right: "Fold of pre" reflects relative PS levels during RS and CS (PS levels during RS and CS divided by the PS during pre-stimulation). **g, i** Since RS and CS were performed in alternation for the same mouse, the effects of adjacent RS and CS sessions were compared. **k–n** PS on a RBR basis across 40 runs for pairs of good performer groups. **k** CHR2 RS vs. EGFP RS; **l** CHR2 CS vs. EGFP CS; **m** CHR2 RS vs. CHR2 CS; **n** EGFP RS vs. EGFP CS. Data are normalized by the mean baseline (pre-stimulation) RBR consistency by session. Horizontal gray lines indicate $p$ values for the group difference from stimulation onset (run 11) to the last post-stimulation run (run 30). All sessions had runs 1–30, while some had more. Individual * indicate a significant difference for a given run. *$p \leq 0.05$, **$p \leq 0.01$, ***$p \leq 0.001$. Error bars and shading represent mean ± sem. See Supplementary Data 1 for exact $n$ and detailed statistical information. Source data are provided as a Source Data file. See also Figs. S9–10.

short 4 m track (Fig. 8d). As in the 10 m NE, the good performers on the 4 m track exhibited higher RBR activity consistency than the poor performers (Fig. S9d–h), and the difference in their activity consistency was most prominent at cues preceding the reward and immediately before the reward (Fig. S9i, j, m). Their difference in calcium response amplitude was also higher at cues but not immediately before the reward in comparison to that in out-cue areas (Fig. S9k, l, n), similar to the observation in the 10 m NE (Fig. S8).

After familiarization with the 4 m track, mice performed at least 30 runs per session on the same track during optogenetics experiments. Within the 30 runs, optogenetic stimulation was administered in the middle 10 runs, allowing for reward-predictive behaviors to be compared between the 10 runs pre-, during-, and post- stimulation (Fig. 8e). For both CS and RS, three sessions were conducted for individual good or poor performers, which were categorized based on their performance during pre-stimulation runs (see Methods). PS and PL of good and poor performers were both comparable to those in imaging mice (Fig. S9o, p). Precise behavioral comparison between the three 10-run epochs in a session was focused on PS, which showed much smaller variation in pre-stimulation epochs across sessions of individual mice compared to PL and, therefore, provided reliable behavioral baselines before stimulation (Fig. S9q).

We compared the effect of three 5 cm RSs with that of three 5 cm CSs at the cues preceding the reward (Fig. 8f). As hypothesized, RS led to a significant reduction in PS, while CS had no effect, in ChR2 good performers on a per-session basis and per-mouse basis (Fig. 8g, h). Stimulations had no effect in EGFP mice (Fig. 8i, j). On a RBR basis, RS (Fig. 8k), but not CS (Fig. 8l), significantly reduced PS in ChR2 relative to EGFP mice. Furthermore, RS in ChR2 mice reduced PS relative to CS (Fig. 8m). PS was reduced in the very first run with RS, slowly recovered post-RS, and reached the pre-stimulation level after around 20 runs (within 4.6 ± 0.6 min) (Fig. 8m). RS and CS comparably had no effect in EGFP mice (Fig. 8n). In the poor performers, neither stimulation paradigm impacted reward-predictive behavior (Fig. S10), suggesting that promoting consistent activity at cues may be insufficient to bolster weak spatial memory.

Together, these results demonstrate that disrupting the consistent MEC map in good performers by RS led to memory impairment, whereas mimicking the consistent map by CS had no effect on memory. This suggests that spatially consistent MEC activity is not only correlated with, but is also necessary for, spatial memory.

## Discussion

Here, we provide several lines of evidence demonstrating the importance of a consistent spatial map in the MEC for effective spatial memory (Fig. 9). First, by tracking calcium activity of MEC neurons for 10 days in 15 mice with different levels of spatial learning in a novel environment (NE), we show that MEC activity gradually became more spatially consistent and then stabilized only in the good but not the poor performers, directly associating a consistent MEC map with effective spatial memory. In addition, the

good performers exhibited increased c-Fos expression upon exposure to NEs, supporting a proper induction of synaptic plasticity that shapes the consistent MEC map during learning and facilitates spatial memory encoding in the MEC. In contrast, c-Fos expression in the poor performers remained high in FEs and NEs, suggesting a hyperactive MEC with dysregulated synaptic plasticity. Finally, optogenetic disruption of spatial activity consistency of the MEC impaired reward-predictive behaviors of the good performers, supporting an essential role of a consistent MEC map for spatial memory.

Furthermore, we provide evidence for the co-existence of a gradually evolving spatial response and stably maintained phase relationship of the grid network during successful learning. Our results suggest that synaptic plasticity from input afferents onto grid cells and rigid recurrent connectivity between grid cells both contribute to constructing a grid map in novel environments.

### Neural dynamics of the MEC during spatial learning

We elucidate neural dynamics of MEC layer 2 during successful spatial learning in two stages: the first day of novel environment exposure and the following nine days of environmental learning.

On day one in the NE, while mice exhibited an immediate decrease in reward-predictive behavior, the MEC showed more active cells, elevated calcium responses, and higher c-Fos expression. This transient network-wide increase in neural response could result from reduced inhibition, increased input amplitudes, elevated intrinsic neural excitability, or changes in neuromodulation[35,57–59], supporting the induction of synaptic plasticity in the MEC upon novelty exposure. Other activity changes included global activity remapping, reduced RBR activity consistency, lower spatial field coverage, and less accurate decoding, suggesting context-specific response of the MEC.

Over the next nine days, the reward-predictive behavior in the NE improved and then stabilized, indicating the establishment of new spatial memory. The improved behavior was coupled with a refinement into a smaller active cell ensemble with a more spatially consistent and environment-specific representation. After 6–7 days of experience in the NE, most activity features reached levels observed in the FE and stabilized. The stabilization of a consistent MEC map agrees with the stable activity of MEC layer 2 neurons in familiar environments[12]. Synaptic plasticity likely contributed to shaping the consistent MEC map, and the plastic changes in the network stabilized after environmental familiarization, as reflected by the shift from high to low c-Fos expression in NEs to FEs. The c-Fos expressions in NEs and FEs are consistent with the encoding of spatial memory in the MEC[50].

We further demonstrated that the consistent MEC map is not only associated with, but also necessary for spatial memory. Previous studies have shown the necessity of rodent MEC in spatial memory through MEC lesion or pharmacological inactivation of layer 2 stellate cells[41,60,61]. Our study, specifically uncovers the disruptive effect of reducing MEC map consistency on spatial memory and, thereby, the necessity of a consistent MEC map for spatial memory.

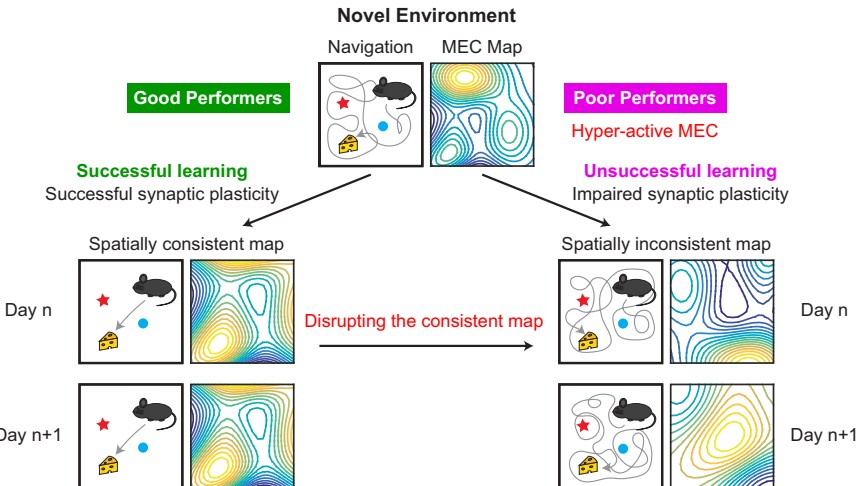

**Fig. 9 | Schematic for the cognitive map of the MEC during spatial learning in a novel environment.** The good performers have successful synaptic plasticity and a spatially consistent MEC map and, therefore, are able to successfully learn the environment. Poor performers have impaired synaptic plasticity (likely due to the hyper-activity in the MEC) and a spatially inconsistent MEC map and, therefore, are unable learn the environment. Disrupting MEC map consistency in good performers leads to reduced spatial navigation performance.

The gradual stabilization of the MEC map during learning is different from the controversial neural dynamics in the hippocampus during experience-dependent navigation. While some studies support the formation of a consistent spatial map in the hippocampus after environmental familiarization[62–68], many other studies showed constantly altered activity of hippocampal neurons in both novel and highly familiar environments[64,65,69–76]. It is worth noting that most of these studies focused on foraging animals without memory demands or did not have a behavioral readout of memory. However, some studies, which demonstrated spatial learning based on behaviors, reported conflicting results: the hippocampus gradually formed spatially stable activity in certain subregions or tasks[64,66,69,70] but showed constantly varied activity in other conditions[69,70]. In addition, it is also unclear whether a stable cell ensemble exists in the hippocampus to encode spatial memory, as a previous study showed that a different subset of CA1 cells was active each day in a familiar environment[76]. The high variability of the hippocampal dynamics raises a possibility that a consistent cognitive map could also be formed in other brain areas. In our study, the experience-dependent increase in MEC map consistency and the necessity of that consistency for spatial memory strongly suggest that the MEC forms a consistent cognitive map and supports spatial memory. The persistent cells, which are constantly active across environments and exhibit environment-specific activity that can be stabilized during multiple days of spatial learning, potentially play a major role in spatial memory encoding. It would be interesting to investigate whether such cells exist in the hippocampus and how their dynamics compare with those in the MEC.

## Dynamics of the grid network during learning

During learning, grid cells gradually increased their spatial tuning consistency while stably maintaining their phase relationships. These observations suggest a role of synaptic plasticity in shaping a spatially consistent grid map in a novel environment and the existence of rigid recurrent connectivity between grid cells to maintain their activity coherency. Therefore, we propose that a model combining synaptic plasticity of feedforward inputs to grid cells and recurrent connectivity between grid cells will better explain the formation of grid cell activity in novel environments.

Theoretically, Hebbian plasticity can act on the grid cell network in at least two ways. First, grid cell activity patterns are shaped by pre-established recurrent connectivity described in CAN models, and Hebbian plasticity from afferents carrying environmental information (e.g., borders and landmarks) onto the grid network aligns grid activity to the environment[26,27]. Alternatively, Hebbian plasticity directly establishes the connectivity to shape grid activity patterns in an environment through specific mechanisms, such as firing rate adaptation[18,19,21,22,25]. Our findings could result from either mechanism. The quickly established and stable landmark-specific tuning of cue cells across environments and throughout learning supports their potential role in providing environmental-specific inputs to the grid network. The existence of feedforward inputs to the grid cell network is also supported by the overall increased grid cell activity correlation during learning in Fig. 5.

In most grid cell models, recurrent connectivity is required to maintain activity coherency between grid cells[15–18,21–24]. The existence of recurrent connectivity between grid cells is currently supported by their stable phase relationships in different environments[42,77] and preserved co-activity patterns during active navigation and sleep[47,48]. Here, we show that despite the significant change in spatial activity patterns of grid cells, their phase relationships and topographical organization remain stable during learning, strongly suggesting that a mature grid network is hardwired through recurrent connectivity that is stable during learning.

## Why could the poor performers not learn?

The key difference between the two performer groups was that after many repetitive visits to previously novel environments, the number of active cells and c-Fos+ cells remained high and the MEC map consistency failed to improve in the poor performers, indicating a hyperactive MEC that could not achieve a steady state after learning. We postulate that the unstable MEC network and weak spatial learning of the poor performers can be attributed to impaired synaptic plasticity, secondary to MEC hyperactivity. This hypothesis is strongly supported by a recent study demonstrating that hyperexcitability of the hippocampus underlies a profound loss of hippocampal LTP in a rat model of psychosis. Interestingly, the hyperexcitable hippocampus exhibited a high basal expression of an immediate early gene, Arc, and Arc expression failed to further increase in novel environments[54], matching our observation in the poor performers that c-Fos level was high in FEs and did not increase in NEs. In agreement with our hypothesis, impaired neural plasticity and cognitive functions when neural networks are hyperactive or hyperexcitable have also been

widely reported in neurological disorders, such as epilepsy, schizophrenia, and Alzheimer's disease[55,78–83]. Future investigations into the circuit mechanism underlying different levels of spatial learning ability will not only advance the understanding of spatial cognition, but also inform on the aberrant mechanisms underlying related neurological disorders.

## Methods

### Animals

All animal procedures were performed in accordance with animal protocol 1524 approved by the Institutional Animal Care and Use Committee (IACUC) at NIH/NINDS. For two-photon imaging experiments, GP5.3 mice[32] (C57BL/6J-Tg(Thy1-GCaMP6f)GP5.3Dkim/J, JAX stock #028280) were used. These included 8 males and 7 females ranging from 4–8 months old at the time when chronic imaging was begun. For histology experiments, TRAP2 mice (Fos^tm2.1(icre/ERT2)Luo^/J, Jackson stock # 030323) were crossed with CAG-Sun1/sfGFP (B6;129-Gt(ROSA)26Sor^tm5(CAG-Sun1/sfGFP)Nat^/J, Jackson stock # 021039) or Ai14 (B6;129S6-Gt(ROSA)26Sor^tm14(CAG-tdTomato)Hze^/J, Jackson stock #007908) mice to generate TRAP2$^{+/-}$; CAG-Sun1/sfGFP$^{+/-}$ (4 males) or TRAP2$^{+/-}$;Ai14$^{+/-}$ (6 males and 10 females) offspring to be used for experiment. Mice ranged from 2.5–4 months of age at time of perfusion. The optogenetics experiments used C57BL/6 J mice ranging from 3.5–5.5 months old at the time of fiber implantation. For the comparison between the three 5 cm CS at cues preceding the reward and three 5 cm RS, 8 ChR2 good performers, 8 ChR2 poor performers, 5 EGFP good performers, and 5 EGFP poor performers were used. Mice were maintained on a reverse 12-hr on/off light schedule with all experiments being performed in the light off period. Animals were housed at a temperature of 70–74 °F and 40–65% humidity.

### Rodent surgeries

All mouse surgeries were performed as follows. Mice were anesthetized using a tabletop laboratory animal anesthesia system (induction: 3% isoflurane, 1 L/min oxygen, maintenance: 0.5–1.5% isoflurane, 0.7 L/min oxygen, VetEquip, 901806) and surgery was performed on a stereotaxic alignment system (Kopf Instruments, 1900). A homeothermic pad and monitoring system (Harvard Apparatus, 50-7220 F) was used to maintain a body temperature of 37 °C. After anesthesia induction, dexamethasone (2 mg/kg, VetOne, 13985-533-03) and saline (500 μL, 0.9% NaCl, McKesson, 0409-4888-50) were administered by intraperitoneal (IP) injection, and slow-release buprenorphine (1 mg/kg, Zoo-Pharm, Buprenorphine SR-LAB) was administered subcutaneously. Enroflox 100 (10 mg/mL, VetOne, 13985-948-10) was used as an antimicrobial wash just after the skull was exposed and just prior to sealing the skull. At the end of surgery, the exposed skull was coated with n-butyl cyanoacrylate tissue adhesive (Vetbond, 3 M, 1469SB). A single-sided steel headplate for head fixation was mounted to the right side of the skull and adhered with dental cement (Metabond, Parkell, S396). Surgeries for histology experiments in Fig. 6 used these methods only.

**Microprism construction.** Microprism construction procedures were similar to those described previously[14,33]. A canula (MicroGroup, 304H11XX) was attached to a circular cover glass (3 mm, Warner Instruments, 64-0720). A right angle microprism coated with aluminum on the hypotenuse (1.5 mm, OptoSigma, RBP3-1.5-8-550), was then attached to the opposite cover glass side. All attachments were performed using UV-curing optical adhesive (ThorLabs, NOA81).

**Microprism implantation surgery.** Microprism implantation procedures were similar to those described previously[14,33]. All insertions were performed on the left hemisphere, aligning with previous observations of more favorable vasculature[14]. A 3 mm craniotomy was performed centered at 3.4 mm lateral to the midline and 0.75 mm posterior to the center of the transverse sinus (approximately 5.4 mm

posterior to the bregma). A durotomy was then performed over the cerebellum. Mannitol (3 g/kg, Millipore Sigma, 63559) was administered by IP prior to the durotomy. The microprism was inserted into the transverse sinus and sealed to the skull with Vetbond. The head plate was then mounted on the skull opposite the craniotomy. Finally, the prism and head plate were adhered to the skull with Metabond.

**Viral injection surgery.** Mice were bilaterally injected with AAV8-hSyn-ChR2(H134R).GFP (Addgene: $3.3 \times 10^{13}$ vg/mL; diluted 6 times in mannitol), AAV5-hSyn-ChR2(H134R)-EYFP (Addgene: $2.4 \times 10^{13}$ vg/mL; diluted 3 times in mannitol), or AAV8-hSyn-EGFP (Addgene: $3.0 \times 10^{13}$ vg/mL; diluted 9 times in mannitol). 100 or 200 nL of virus was pressure-injected through a glass micropipette at each injection site at a rate of 100 nl/min. On each hemisphere, mice were injected at 2 sites in the MEC (0.77 mm anterior to the transverse sinus, 3 mm lateral to bregma, 1.79 mm from the surface of the brain; 0.6 mm anterior to the transverse sinus, 3.36 mm lateral to the bregma, 1.42 mm from the surface of the brain) with mice heads tilted up 18°.

**Fiber implantation.** At least 3 weeks following the viral injection, mice were chronically implanted bilaterally with Lambda fibers (Plexon) at 0.3 mm anterior to the transverse sinus, 3.2 mm lateral to bregma, and inserted to a depth of 2.5 mm from the brain surface as described previously[84]. Fiber dimensions were as follows: 0.66 NA, 3.0 mm total length (1.0 mm implant length; 2.0 mm active length; 200/230 μm core fiber). Following the fiber insertion, a thin layer of Vetbond was first applied followed by a thick layer of Metabond to cover the exposed skull.

### Virtual reality setup

For all behavioral experiments, a customized virtual reality (VR) setup was used, which projects a one-dimensional (1D) virtual environment based on the running of a mouse, similar to that described previously[14]. Mice were head-fixed onto an air-supported polystyrene ball (8" diameter, Smoothfoam) using the mounted head plate. The ball rotated on an axle, allowing only forward and backward rotation. The virtual environment was projected onto a hemispherical dome filling the visual field of mice (270° projection). An optical flow sensor (Paialu, paiModule_10210) with infrared LEDs (DigiKey, 365-1056-ND) was used to measure the rotation of the ball and thereby control the motion of the virtual environment. The optical flow sensor output to an Arduino board (Newark, A000062), which transduced the motion signal to the computer controlling the virtual reality. An approximately 4 μl water reward was provided via a lick tube at fixed locations (1 or 2 reward locations) in a given environment using a solenoid. A lick sensor connected to both the lick tube and head plate holder was used to detect mouse licking. A mouse licking the lick tube created a closed circuit between the lick sensor, the lick tube, the mouse (from the tongue to the skull), the headplate (which directly contacts the skull), and the head plate holder. The solenoid and lick sensor were controlled using a Multifunction I/O DAQ (National Instruments, PCI-6229). The virtual environments were generated and projected using ViRMEn software (Princeton, version 2016-02-12)[85]. Imaging and behavior data were synchronized by recording a voltage signal of behavioral parameters from the VR system using the DAQ. ViRMEn environments were updated at 60 Hz. The DAQ input/output rate was 1 kHz. The synchronization voltage signal was updated at 20 kHz. Final behavioral outputs were matched to the imaging frame rate (30 Hz, see below) for synchronization.

Environments were colored blue and projected through a blue wratten filter (Kodak, 53-700) to reduce contamination of the imaging path with projected light. Virtual environments were 1D linear tracks with patterned walls and patterned visual cues at fixed locations (Figs. 2b, S1a, S1b). At the end of the track, mice were immediately teleported to the start of the track. Imaging experiments used a

1000 cm track. Optogenetic experiments used a 400 cm track. Histology experiments used a 400 cm (4 mice) or 1000 cm (16 mice) track.

## Behavior

**Training.** Mice were allowed to recover for 5 days post-surgery (prism implantation, virus injection, or optogenetic fiber implantation) and were then put on water restriction, receiving 1 ml water per day. After approximately 3 days of water restriction, mice were trained daily in VR. Training experience is summarized in Table S1.

**Imaging mice training.** For imaging mice, some mice were trained on a 1 m track to encourage running (8 mice: 3 out of 15 mice on 10 m tracks and 5 out of 6 mice on 4 m tracks) and/or were first trained on other VR tracks for other experiments (8 mice: 2 out of 15 mice on 10 m tracks and all 6 mice on 4 m tracks). All mice were further trained on a 10 m or 4 m track until familiarization (familiar environment, FE), as measured by consistent running (>40 trials per hour) and stable significant anticipation of reward locations for 3 days (less than 5% change in predictive licking and slowing, see "Predictive Licking" and "Predictive Slowing" for quantification). The mice were then switched to the novel environment (NE, the same length as the corresponding FE) for spatial learning experiments.

The experience of all mice in FEs was comparable (Table S1) and was not correlated with behavioral performance, as shown in Fig. S1i. The prior experience of the 8 mice in other experiments did not affect their performance in NEs, as the two 10 m imaging mice with prior experience (49 ± 19 days) were categorized as one good and one poor performer, and the six 4 m imaging mice with prior experience (74 ± 9 days) were categorized as four good and two poor performers, the ratio of which (4 good: 2 poor) was similar to that of 10 m imaging mice (11 good: 4 poor).

**Histology mice training.** Initial training of histology mice was performed on a 400 cm (4 mice) or 1000 cm (16 mice) track, which was used as the FE. After 4–6 weeks of training in FE, half of the mice were exposed to NEs, while the remaining half were again exposed to FEs. The NE had the same length track, but the cues had different locations and patterns and the reward locations were different. After being allowed to explore the FE or NE for 1 h, the mouse was perfused for immunohistochemistry.

**Optogenetic mice training.** After recovery from viral injection, optogenetic mice were first trained on a 1 m track, which motivated them to run, and then trained directly on the 4 m track (Fig. 8d) used later during stimulation. During training, some mice were temporarily trained on shorter tracks (100–200 cm) to encourage running. Once half of the mice in a cohort (~10 mice/cohort) reached the criterion for good performers on the 4 m track (see "Spatial Learning Criteria: optogenetic mice" for details), fiber implantation was conducted on all mice in the cohort (5–8 weeks after viral injection). After recovery from fiber implantation surgery, the mice were trained on the same familiar 4 m track again for optogenetics experiments

Mice used in histology and optogenetic experiments had more experience (in training days and in laps traversed) in their respective FEs than imaging mice had in NEs by the end of the 10 experimental days, as shown in red numbers in Table S1. Therefore, FEs used in the histology and optogenetics experiments corresponded to times after the 10 days of learning in imaging experiments.

**Predictive licking.** Predictive licking (PL) was measured as the percentage of licks that occurred within a specific distance threshold (20 cm, except for spatial learning criteria as below) prior to the reward delivery relative to all other locations (excluding 30 cm after reward). When determining good and poor performers, various PL zone sizes were used, as described below.

**Predictive slowing.** To calculate predictive slowing (PS), for each run, the velocity changes between each data point were calculated along the track. The pre-reward acceleration value was calculated as the mean velocity changes ($\Delta v$) within a given window size (distance threshold) before the reward (for example 690 cm to 890 cm for a 200 cm window size and a reward location at 890 cm). The rest of the track was analyzed using a rolling average of $\Delta v$ for the same window size at 1 cm intervals, generating a series of comparison acceleration values. The slowdown percentile for a given lap was the percentile of the pre-reward acceleration within the comparison acceleration values from the rest of the track, such that a higher percentile signified a more negative acceleration (i.e., deceleration) and thus better PS. The rolling averages calculated for the comparison values excluded any window intersecting the edge of the track or areas close to the reward (from 90 cm before to 30 cm after reward delivery) to avoid edge effects and reward related behavior, respectively. Thus, for a 200 cm window size, the comparison windows ranged from 0–200 cm to 600–800 cm on the 10 m novel environment track. Depending on the distance threshold used, these windows may have some overlap with the pre-reward window. This overlap was allowed to increase the percent of the track covered when calculating comparison values. For example, in the novel environment with a 200 cm window, the last comparison window would begin at 490 cm if no overlap is allowed, excluding a large proportion of the track.

Except as noted for spatial learning criteria below, the distance thresholds used were 150 cm for 10 m tracks and 70 cm for 4 m tracks, which corresponded to pre-reward windows of 385–535 cm for 10 m familiar environment, 740–890 cm for 10 m novel environment, and 296–366 cm for 4 m track. Note that for the 10 m familiar environment, only the second reward location was used to compute the slowdown percentile and only 50–700 cm of the track were included for calculation to exclude the first reward and because several mice started slowing at the end of the track in anticipation of the first reward.

The slowdown percentile of a session was computed as the average slowdown percentile of all the runs within the session.

**Slowing events.** Slowing events were calculated on a run-by-run basis, ignoring the beginning and final incomplete runs. First, the rolling average of velocity (70 time point window, ~1.2 s at 60 Hz) was calculated for a given run. Then local velocity minima were found using the MATLAB "findpeaks" function, defined here as slowing valleys. The input parameters were set such that slowing valleys were at least 5 time-bins apart and the minimum speed difference between a slowing valley and the adjacent peaks before and after the valley was at least 2 cm/s. The number of slowing events for a given run was defined as the number of slowing valleys identified.

**Spatial learning criteria: imaging mice.** Mice were categorized as good and poor performers based on spatial learning ability as measured by both PS and PL, which both correlate with familiarity with an environment[34]. PS and PL were calculated with different parameter settings (PS: 50, 100, 150 and 200 cm distance threshold; PL: 20, 35, 50 and 75 cm distance threshold) for all 10 days in the 1000 cm NE and averaged across days, generating 8 values (4 from PS and 4 from PL) per mouse. All 15 mice were ranked based on each set of values from high to low. The mice that were consistently ranked in the lower half of mice (i.e., with low performance values) for all 8 sets of values regardless of the chosen parameters were categorized as poor performers. The remaining mice were categorized as good performers.

**Spatial learning criteria: histology mice.** The good and poor performers in histology mice were determined based on their PS and PL in

FEs. Since both 10 m and 4 m tracks were used as FEs for the histology mice, PS on the 10 m and 4 m tracks was calculated using 150 cm and 70 cm distance thresholds, respectively. PL on both 10 m and 4 m tracks was calculated using a 20 cm distance threshold. The PSs and PLs of each mouse in the last 6 days of FEs were further averaged as avgPS_hist and avgPL_hist for the mouse, respectively.

The good and poor performers were then determined based on the thresholds developed from imaging mice: the PS and PL values of the four imaging poor performers in the last 6 days (days 5–10 in NE) of learning were first averaged as avgPS_img, avgPL_img, respectively. The thresholds for PS and PL were determined as the maximal values across the poor performers of avgPS_img (65.7th percentile) and avg-PL_img (22.2%), respectively. The histology mice with both avgPS_hist and avgPL_hist below the corresponding thresholds were assigned as poor performers. All other mice were assigned as good performers. The good and poor performers in histology mice categorized using this method showed comparable PS and PL as those in imaging mice (Fig. 6b, c).

**Spatial learning criteria: optogenetics mice.** PS and PL of the optogenetics mice on the 4 m track were calculated using 70 cm and 20 cm distance thresholds, respectively. Since optogenetics experiments were conducted on a session-by-session basis (one session per day) without averaging multiple sessions/days as what was done for imaging and histology mice, we needed to classify the good and poor performance of the mice per session using a reliable parameter. Therefore, PS was used to indicate mouse performance, as it showed much smaller variations than PL across individual sessions of the same mouse (Fig. S9q). As described above ("Spatial Learning Criteria: histology mice"), a mouse with PS below 65.7th percentile (PS threshold) in a session was assigned as a poor performer in that session. The good and poor performers categorized using this method showed comparable PS and PL with the imaging mice (Fig. S9o, p). To be consistent, this classification method was applied to all steps of the optogenetics experiments.

During early training (after viral injection and before fiber implantation), when PS of half the mice in the cohort (~10 mice/cohort) reached the PS threshold for at least two consecutive days (5–8 weeks after viral injection), fiber implantation was conducted for all mice in the cohort. One mouse that never learned to run was removed from the experiment.

After recovery from the implantation surgery, mice were trained again until their PS was stable (consistently above or below PS threshold for at least 2 days) in the same 4 m track (normally 7–10 days of training). Optogenetics experiments were then conducted on a daily basis with only one CS or one RS session performed per day. For stimulation (three 5 cm CS at cues versus three 5 cm RS), 3 CS sessions and 3 RS sessions were conducted in alternation for each mouse (one stimulation session per day for 6 total days) so that the effects of adjacent CS and RS sessions could be compared (three pairs of comparison per mouse, as in Fig. 8g, "compare"). As described above, the sessions with PS above and below the PS threshold were assigned to good and poor performances, which should largely correspond to high and low spatial consistency of MEC activity, as demonstrated by our imaging data on an individual-session basis (Figs. S4a, S9g). For the comparison between the three 5 cm CS at cues preceding the reward and three 5 cm RS, 8 (4 male, 4 female) ChR2 good performers, 8 (5 male, 3 female) ChR2 poor performers, 5 (2 male, 3 female) EGFP good performers, and 5 (2 male, 3 female) EGFP poor performers were used.

**Behavior fitting.** To determine when the good-performing mice achieved stabilized behavioral performance, PS and PL calculated with different parameter settings were fitted with a one phase association exponential function (Prism 9.3.1, GraphPad), where y is the PS (or PL), and x is day. The start points of fitted curves were set to be 0% for PL

(no predictive licking) and the 50th percentile for PS (the random level of PS). Once the values for the plateau and K, a rate constant, were derived, the number of days required to achieve 95% of the plateau was calculated.

**Visual discrimination task.** Our visual discrimination task was based on established visual discrimination protocols but adapted for head-fixed mice in our virtual reality setup[86,87]. In each session, mice experienced 98–379 individual trials. For analysis, the first 98 trials per session were analyzed. Each trial began with a 3–5 s (uniformly distributed at random) inter-trial interval during which the VR projection showed a black screen. A patterned wall moving at 10 cm/s in the direction of mouse motion was then shown for 6 s. Mouse running speed did not affect wall motion. Each grouping of three trials displayed the three cue patterns on the wall in a randomly permuted order. Wall patterns included the target cue (vertical stripes, with reward) and the non-target cues (diagonal stripes and dots, without reward). All three patterns were used as spatial landmark patterns in the 10 m NE. The test zone was 4 s in duration beginning 1 s after the start of the cue presentation. During target cue trials, if a mouse licked during the test zone, it immediately received a 4 μl water reward. If no lick occurred before the end of the test zone, a water reward was given to reinforce association of the target cue with the reward. A mouse could only receive 1 water reward per trial. No reward was given during the presentation of non-target cues regardless of mouse licking, and no punishment was given in response to incorrect licking. All licks during a session were recorded, but only those during test zones were used for the calculation of the discrimination factor. The discrimination factor (d-prime) was calculated according to the formula: d-prime = norminv(HR)-norminv(FA); where HR=Hit Rate or the fraction of target cue trials with licking during test zone, and FA=False Alarm Rate or fraction of non-target cue trials with licking during test zone[88]. A threshold of d-prime>1 was used to denote successful visual discrimination[86,88].

## Two-photon imaging

Imaging was performed using an Ultima 2Pplus microscope (Bruker) configured with the above VR setup. A tunable laser (Coherent, Chameleon Discovery NX) set to a 920 nm excitation wavelength was used. Laser scanning was performed using a resonant-galvo scanner (Cambridge Technology, CRS8K). GCaMP fluorescence was isolated using a bandpass emission filter (525/25 nm) and detected using GaAsP photomultiplier tubes (Hamamatsu, H10770PB). A 16x water-immersion objective (Nikon, MRP07220) was used with ultrasound transmission gel (Sonigel, refractive index: 1.3359[89]; Mettler Electronics, 1844) as the immersion media.

The anterior-posterior (AP) and the medial-lateral (ML) angle of the prism (i.e., the angle of the surface of the prism along to the AP or ML direction of the mouse) relative to the head-fixed position of the mouse were measured prior to the first imaging session. The head plate holder and rotatable objective angles were set daily to align the objective with the prism in the AP and ML direction, respectively, such that the objective was parallel to the prism surface. Black rubber tubing was wrapped around the objective and imaging window to prevent light leakage into the objective.

Microscope control and image acquisition were performed using Prairie View software (Bruker, version 5.5). Raw data was converted to images using the Bruker Image-Block Ripping Utility. Imaging was performed at 30 Hz with 512 × 512 resolution. Average beam power at the front of the objective was typically 70–115 mW. Imaging and behavior data were synchronized as described above.

## Image processing

Imaging data was down-sampled by a factor of three by taking the average of each consecutive block of 3 frames and processed as

previously described using published MATLAB scripts[33]. Motion correction was performed using cross-correlation based, rigid motion correction. Identification of regions of interest (ROIs, active cells) with correlated fluorescence changes was performed using principal component analysis combined with independent component analysis[90]. The fluorescence time course of individual ROIs was then extracted. The fractional change in fluorescence with respect to baseline ($\Delta F/F$) was calculated as $(F(t) - F0(t)) / F0(t)$[14]. For each cell, significant calcium transients were identified using amplitude and duration thresholds, such that the false-positive rate of significant transient identification was 1%[13]. A final $\Delta F/F$ including only the significant calcium transients was used for all further analysis.

The mean $\Delta F/F$ (significant transients only, as mentioned above) for a cell was calculated as a function of position along the track in 5 cm bins. Data points when the mouse was moving below a speed threshold were excluded from this analysis. The speed threshold was calculated by generating a 100-point histogram of all instantaneous velocities greater than 0 and taking the value twice the center of the first bin (approximately 1% of max positive speed).

To remove artifactual ROIs occasionally caused by light leak, ROIs with an extreme non-circularity (ROIs were the ratio of the major to minor axis of the ellipse with the same normalized second central moments as the region was greater than 3.3) were removed from further analysis.

**Cell alignment**. All imaging sessions for a given mouse were aligned pairwise, as previously described[91], to identify common cell pairs between each pair of sessions. Pairwise alignments were combined to generate the full set of possible cell alignments for all imaging days. These possible alignments were manually checked to determine the set of cells identified in all imaging sessions. Similarly, cells tracked for fewer than 11 days were identified by utilizing the pairwise alignments for all imaging days. Due to the pairwise nature of the alignment, some cells from some days appeared in multiple similar alignment groupings. Such groupings were combined to identify the first day a cell was tracked. The number of days a cell was tracked was calculated as the maximal tracked days in all alignment groupings.

**Data analysis**
**Stellate/pyramidal cell classification**. To identify the total cell number of cells in a FOV regardless of activity, cells were manually traced from motion corrected maximal projections of each FOV. Cells were classified as stellate cells or pyramidal cells based on the bimodal distribution of their long-axis diameters as previously described[33]. The valley between the two peaks of cell diameters was 16.6 μm. To increase the confidence of cell classification, cells with diameters smaller than 15.6 μm and larger than 17.6 μm (16.6 ± 1 μm) were classified as pyramidal and stellate cells, respectively.

**Matching manually identified cells to active cells**. To identify which manually identified cells corresponded to which active cells, the closest active cell to each manually identified cell was calculated using the centroid-to-centroid distance. A given pair of manual and active cells was considered to be the same cell if this centroid-to-centroid distance was less than 10 μm.

**Anatomical clustering analysis**. Anatomical cell clustering was performed similar to previous analysis[40]. c01 cells were all cells tracked on both day 0 and day 1 (Fig. 2k). "Other" cells were all cells that were tracked on only one of day 0 and 1 (a0 and a1, Fig. 2k), as well as all cells that were first tracked after day 1 (a2-a10, Fig. 2k). To avoid edge effects caused by unequal coverage of FOVs, only "other" cells that mapped to the region covered by both day 0 and 1 were included in the "other" category. To avoid artifacts caused by unequal numbers of cells between groups or across FOVs, the minimum number of cells for

either group in any FOV was calculated ($n = 39$ cells). When performing the following analysis, random subsets of 39 cells were selected from both the common and other cells within a FOV. The analysis was averaged across 1000 such subset pairs. To calculate the nearest-neighbor (NN) distances for a given FOV, for each cell in a group (c01 cells or other cells), the centroid-to-centroid distance was averaged for the N-nearest cells in that same group ($N = 1-38$). These calculated NN-distances were then averaged for each value of N to determine the average NN-distance for that group across the FOV.

**Inter-day activity correlation**. The activity matrix correlation for a given set of cells in a particular imaging session was calculated as follows: the spatially binned mean $\Delta F/F$ for each cell was averaged across all runs along the track, generating a 1D array. The calculated array for each cell was concatenated to generate a single matrix (size = total number of cells by number of spatial bins) such that the activity of a given tracked cell was in the same row of the activity matrix for all days. The day-to-day activity matrix correlation was the 2D correlation between the calculated matrices for two consecutive days when the cell activity was placed in the same order in both matrices.

The activity correlation for individual cells was calculated similarly to above, except the mean $\Delta F/F$ averaged across runs was not concatenated with other cells. Instead, day-to-day activity correlation for a given cell was the 1D correlation between the generated 1D arrays for that cell on two consecutive days.

**Spatial field distribution**. To calculate significant spatial firing fields, regions of the track with significantly consistent activity, the mean $\Delta F/F$ was compared to shuffles of the original $\Delta F/F$ as described previously[33]. Each shuffle was calculated such that the original spatial position of each time point was preserved, but the $\Delta F/F$ was shuffled by bisecting the full $\Delta F/F$ time course at a random time point and swapping the order of the resulting halves. The mean $\Delta F/F$ of the shuffle was then calculated as described above. A spatial field was defined as a region of at least 3 consecutive 5 cm bins (except that the fields at the beginning and end of the track could have 2 bins) that had a mean $\Delta F/F$ higher than 80% of 1000 shuffles at the corresponding bins. Additionally, at least 10% of runs were required to have a significant calcium transient in the spatial field.

The spatial field distribution includes all spatial bins contained within all spatial fields for a given imaging session. For the activity of each cell on one day, a bin vector (200 elements corresponding to 200 bins on 1000 cm track) was generated by using ones and zeros to indicate whether individual spatial bins were in a spatial field or not, respectively. The bin vectors of all cells were grouped in a 2D matrix (M, size = total number of cells by number of spatial bins), representing the spatial field distribution of all cells on a given day. To calculate the significance between the two spatial field distributions before (days 1–2) and after learning (days 7–10), we averaged the Ms on days 1 and 2 (M1), and days 7–10 (M2), and compared M1 and M2 on a bin-by-bin basis (columns of M1 and M2).

**Intra-day run-by-run (RBR) activity consistency**. The run-by-run consistency for a given cell in a particular imaging session was calculated as previously described[45]. The spatially binned mean $\Delta F/F$ for each run along the track was correlated with that of every other run. The average of these correlations is the run-by-run consistency value for a given cell.

**Position decoding**. Position decoding was performed by separating the imaging data into template data (odd runs) and testing data (even runs) as described previously[45]. The spatially binned mean $\Delta F/F$ for the template data runs were averaged across runs and concatenated for all cells to generate a template matrix (size = total number of cells by number of spatial bins). The $\Delta F/F$ of all cells at each time was then

correlated to each spatial bin of the averaged template data matrix. The decoded position for a given time point was the spatial position of the template bin that gave the highest correlation. Decoding error for each time point was calculated as the absolute difference between the decoded spatial position and the true position at that time point. For calculating correct decoding percentage, a time point was considered correctly decoded if the decoding error was less than 10 cm. Due to unstable running of behavior at the beginning and end of the track, the data in the first and last 150 cm were removed[45]. The decoding was performed using either all cells or 50 randomly chosen cells. When 50 cell were used, two random selections of 50 cells were made for each FOV and the averaged the value was used for the decoding performance of the FOV.

**Speed exclusion criteria.** For determining the subset of the good performers with comparable speed to the poor performers, the average speed of each mouse on individual days was calculated. If the maximum average speed of a good performer was higher than the maximum average speed of any poor performer, it was excluded from this subset of mice.

**Grid/cue cell classification.** Grid cells were identified as described previously based on the features of spatial fields[33]. Cue scores were calculated as previously described[29]. Cue score thresholds were separately calculated for left and right cue cells as the 95th percentile of the left and right cue scores for 200 shuffles of each cell from all mice and imaging sessions in a given environment, respectively. In the rare cases when a cell was above the left and right cue score threshold, the cell was classified as the side with the higher cue score. Left and right cue cells were combined for all analyses. A cell that met the criteria for both cue cells and grid cells was only classified as a cue cell. Aligned cells were declared true cue cells or grid cells only if they were identified as such in more than half the imaging sessions they were tracked for. For example, a cell tracked for all 11 days was only classified as a true cue or grid cell if it was identified as such on 6 or more days.

**Grid scale/grid field spacing.** The grid scale of a particular grid cell was calculated as the smallest distance between the centers of any two adjacent significant spatial firing fields, which were calculated as above.

**Cue cell statistics.** The spatial shift and response amplitude for cue cells were calculated on a run-by-run basis from the spatially binned mean ΔF/F for each cue individually. The spatial shift was the number of bins to shift cue cell activity in a particular run so that the shifted activity had the maximal correlation with the cue template. The spatial shift was then converted to centimeters. The response amplitude was the peak value of mean ΔF/F of the response to individual cues. The standard deviation of each measure was calculated for each cue separately and averaged together to generate the standard deviation value for a given cell. The variation in response amplitude for each cell was normalized by the mean ΔF/F of the cell.

**Grid module identification.** Grid modules were identified based on field spacing and width of grid cells as described previously[33]. Throughout the 11 days, the field spacing and width of a cell were determined as the minimal spacing (as described above) and maximal field width that appeared on at least two days, respectively. Within each FOV, field spacing and widths of all cells were clustered and cells in different modules were assigned according to the clustering. The grid module with the smallest grid spacing in each mouse was further identified based on the module spacings in all FOVs of the same mouse. Generally, each mouse had 2–3 FOVs, which spread across the first 1 mm along the dorsal and ventral axis of the MEC and contained 2–3

modules. The spacing of the smallest module was generally around 50 cm.

**Pairwise activity correlations and day correlation of grid cells.** The pairwise activity correlations of grid cells were calculated using the ΔF/F containing only significant transients. To be consistent across cells and behavioral sessions, the first 4000 data points (~6.7 minutes during behavior) of the ΔF/F were used for all calculations. The pairwise activity correlations of grid cells were calculated using the significant transient – only ΔF/F binned as a function of track positions. Day correlations were further calculated by correlating the pairwise activity correlations on adjacent days. Shuffled correlations were made by randomly permuting pairwise correlations 200 times among grid cell pairs in individual FOVs on individual days and recalculating Day correlation using the permuted data. For pairwise correlation as a function of pairwise distance, data are grouped in 30 μm bins. The adjusted Pairwise corr. on each day was calculated as the original correlation subtracted by the mean Pairwise corr. of grid cell pairs at all distances on the same day.

**RBR activity consistency along the track.** RBR activity consistency along the track was calculated by calculating the RBR activity correlation (as in "Intra-day run-by-run (RBR) Activity Consistency") within a rolling window of 5 spatial bins. For a 1000 cm track with 200 spatial bins, the correlations were calculated for areas within bins 1 to 5, 2 to 6, 3 to 7, …196 to 200.

## Optogenetics

**In vivo electrophysiology.** To confirm the efficacy of the ChR2, after approximately 4 weeks of viral injection with AAV8-hSyn-ChR2(H134R)-GFP, multi-unit recordings were performed using customized tungsten electrodes (California fine wire[92]) and a 32-channel amplifier chip (Intan technologies, RHD2132). Mice were anesthetized with 0.5–1.5% isoflurane and placed into a stereotaxic frame to make a craniotomy to insert an optrode (tungsten electrodes glued to an optical fiber (Thor labs, 0.39 NA, Ø200 μm cut and coupled to ferrule in-house) into the MEC (0.15 mm anterior to the transverse sinus, 3.24 mm lateral to the bregma, 1.5–2.2 mm from the surface of the brain). The location of the optrode was confirmed by advancing the electrode toward MEC and monitoring for light induced changes in the multi-unit activity. Spikes were acquired using an Open Ephys (Open Ephys GUI, version 0.5.5.2) recording system at 30 kHz. Before experiments, a blue LED module (465 nm, PlexBright LD-1 Single Channel LED Driver from Plexon) was coupled to the optrode fiber, and the blue light at approximately 4–5 mW was used for electrophysiology (measured at the tip of the fiber using Sphere Photodiode Power Sensor; Thorlabs S142C). During recordings, light pulses were 2 ms long and were delivered at 10 Hz for 1 s, followed by 10 s inter-train intervals, and repeated over 3 times at each depth. The LED was triggered using the Pulse Pal 2 (Sanworks) controlled by custom MATLAB scripts.

**Multi-unit spike detection.** High frequency voltage traces were first extracted using a bandpass filter with cutoff frequencies between 250 and 8000 Hz[93]. Spikes were then defined as contiguous voltage deflections that were larger than 3 standard deviations from the mean.

**Optical stimulation during behavior.** Behavior experiments for optical stimulation were performed as described above using a 4 m track. Before experiments, a blue LED module was coupled to the Lambda fibers, and the blue light was used at ~8–9 mW. Each stimulation session began with 10 no-stimulation runs, followed by 10 consecutive stimulation runs and then switched back to 10 no-stimulation runs (after the 5 excluded runs, see below). For each stimulation run, specific locations along the 4 m track (excluding the first 5 cm and the

track area after the reward) were selected for stimulation, as indicated in Fig. 8f. When the mouse location was within the stimulation zone and the speed was above 1 cm/s, a 10 Hz pulse train (2 ms duration; controlled by Pulse Pal 2, Sanworks) of 465 nm light was delivered. If the mouse was within the stimulation zone and paused (speed <1 cm/s), the pulse train was not delivered until the mouse moved again (speed >1 cm/s). The connection area between the fiber and optic patch-cord was covered by black aluminum foil (BKF12, Thorlabs) to minimize potential visual distraction caused by blue light leakage. The first five runs of each behavioral session were excluded from behavioral analysis, as some mice required a short period of time to warm up.

## Immunohistochemistry

Mice were anesthetized with a ketamine (200 mg/kg, VetOne, 13985-584-10) and xylazine (20 mg/kg, VetOne. 13985-612-50) cocktail and were transcardially perfused with 4% paraformaldehyde (PFA, Electron Microscopy Sciences,15713) in phosphate buffer solution (PBS, Corning, 46-013-CM). Brain tissues were dissected and fixed in 4% PFA in PBS overnight at 4 °C. Sagittal slices (40 μm thick) were prepared using a VT1200S vibratome (Leica Biosystems). Slices were washed in PBS (3 × 10 min), and then blocked with 10% bovine serum albumin (BSA, Millipore Sigma, A3294), 0.5% triton X-100 (Millipore Sigma, T9284), and PBS for 1 h at room temperature. Primary and secondary antibodies were diluted in 2% BSA, 0.4% triton, and PBS. Slices were incubated in diluted primary antibodies overnight at 4 °C, then washed in PBS (3 × 10 min). Slices were incubated in diluted secondary antibodies (1:500 dilution) for 1.5–2 h at room temperature, then washed in PBS (3 × 10 min) and mounted with mounting medium (Vector Laboratories, H-1000-10). Images were collected using a Zeiss 880 spectra confocal. Primary antibodies used include rabbit anti-c-Fos (1:2000, Cell Signaling Technology, 2250 S), mouse monoclonal IgG1 anti-Reelin (1:1000, Abcam, AB78540), mouse monoclonal IgG2a anti-Calbindin (1:3000, Abcam, AB75524), and mouse monoclonal IgG2a anti-GAD67 (1:2000, Millipore Sigma, MAB5406). Secondary antibodies included Alexa 488 conjugated goat anti-rabbit antibody (Invitrogen, A32731), Alexa 568 conjugated goat anti-mouse IgG1 antibody (Invitrogen, A21124), Alexa 568 conjugated goat anti-mouse IgG2a antibody (Invitrogen, A21134), and Alexa 647 conjugated goat anti-mouse IgG1 antibody (Invitrogen, A21240). Nissl (1:100, Invitrogen, N-21479) staining was performed on individual slices to label neurons.

**Quantification of histology.** Each cell was selected using "Cell Magic Wand Tool" and its intensity of c-Fos, Calbindin and Gad67 immunostaining was measured in Fiji. Afterward, the intensity distribution for each brain slice was plotted using "ksdensity" built-in function in MATLAB with width setting to get reasonable distribution (width for c-Fos, Calbindin and Gad67 is 400–500, 1000–2000, and 1000, respectively). The cut-off for c-Fos, Calbindin and Gad67 are the first valley after 8000, 5000, and 20000, respectively.

## General data analysis and statistics

Image processing was performed using previously published MATLAB (MathWorks, version R2015aSP1) codes as cited above. Data analysis was performed using ImageJ (Fiji, version 1.53q), MATLAB (MathWorks, versions R2015aSP1 and 2020a), and GraphPad Prism (version 9.3.1). Linear correlations and the corresponding $r$ and $p$ values were calculated using a two-tailed Pearson's linear correlation coefficient. Other significance values were calculated using Student's $t$ test, Student's paired $t$ test, or ANOVA tests (two- or three-way ANOVA, or repeated measures ANOVA; sphericity is assumed for all tests). Where appropriate, multiple pairwise comparisons were performed following ANOVA tests using Student's $t$ test or Student's paired $t$ test. $P$ values were adjusted for multiple comparisons using the Bonferroni–Holm

method as noted. For anatomical clustering, pairwise comparisons were performed using the two-sided Mann–Whitney $U$ test. Two-tailed tests were used for all analyses expect for analysis of pairwise correlation of grid cells with respect to distance. One-tailed tests were performed for these tests in the direction predicted by the distance between cells[13]. $P$ values less than 0.05 were considered significant (* < 0.05, ** <0.01, *** <0.001). All figures show mean and standard error, except where noted. Detailed statistical information for all figures can be found in Supplementary Data 1.

## Reporting summary

Further information on research design is available in the Nature Portfolio Reporting Summary linked to this article.

## Data availability

Raw data are extremely large and not feasible for upload to an online repository but are available upon request to yi.gu@nih.gov. Processed source data for all figures and associated statistical analysis are provided with the paper. Source data are provided with this paper.

## Code availability

Custom MATLAB scripts used for data analysis are available on GitHub (https://github.com/GuLab-NIH/MaloneNatComm2024).

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

## Acknowledgements

We thank all colleagues in the Gu laboratory for supporting the work, the Section on Instrumentation at National Institute of Mental Health for help with building virtual reality setup, Dr. Chris McBain, Dr. Lorna Role, Dr. Katherine Roche, Dr. Jeff Diamond, Dr. Edward Giniger, Dr. Mark Wagner, Dr. Guangfu Wang, and Dr. Bailu Si for insightful discussion of the data and constructive suggestions on the manuscript, Dr. Sean Bradley in Rodent Behavioral Core at National Institute of Mental Health for helping design the visual discrimination task, the NINDS Light Imaging Core Facility, and Dr. Jim Heys at the University of Utah for helping with optogenetics experiment. This work was supported by the NIH/NINDS Intramural Research Program (to Y.G.).

## Author contributions

Conceptualization and Methodology, T.J.M., N.-W.T., Y.M., Y.G.; Investigation, T.J.M., N.-W.T., Y.M., L.C., S.L., G.W., D.N., K.Z., M.V.N., J.T., and Y.G.; Software, T.J.M., N.-W.T., Y.G.; Formal Analysis, T.J.M., N.-W.T., Y.M., Y.G.; Writing – Original Draft, T.J.M., N.-W.T., and Y.G.; Writing – Reviewing & Editing, T.J.M., N.-W.T, D.A.K., and Y.G; Resources, J.A.G. and Y.G.; Supervision, J.A.G., D.A.K., and Y.G.; Funding Acquisition, Y.G.

## Competing interests

The authors declare no competing interests.
