## [Peer Review File · Nature Communications]

A consistent map in the medial entorhinal cortex supports spatial memoryEditorial Note: This manuscript has been previously reviewed at another journal that is not operating a transparent peer review scheme. This document only contains reviewer comments and rebuttal letters for versions considered at *Nature Communications*.

REVIEWERS' COMMENTS

Reviewer #1 (Remarks to the Author):

The revised version is substantially improved and addresses all of my previous comments. The description of the findings and of the claims that can be made based on these findings is now clear and transparent. Below are minor points that could be further clarified.

Line 331. The subtitle 'Good but not poor performers have increased c-Fos expression upon novelty exposure' does not capture the finding that the failure to find an increased c-Fos expression in poor performers can be explained by the already high expression in FE. The subtitle should be rephrased so that it is not misinterpreted as implying that c-Fos expression was consistently low in poor performers.

Line 368-371. '... would mimic the pattern of the MEC map ...' could be misinterpreted as implying that the stimulation reproduces the original MEC map. What is perhaps meant here is that the stimulation mimics that the MEC map is stable, which could be more directly stated.

Figure S5, panels a-d. It is not clear what the comparison group is for the 'Percentage' labels on the vertical axis. What does 1005 correspond to. For example, in a, is it all recorded cells, all grid cells that were identified on a single recording day?

Reviewer #3 (Remarks to the Author):

The authors have performed extensive work in this revision and I have no further concerns. This represents an important set of experiments showing how long-term grid cell stability may relate to navigation.

Response to Review Comments:

Below are our revisions of the manuscript according to the referees' comments from the third review.

Reviewer #1 (Remarks to the Author):

The revised version is substantially improved and addresses all of my previous comments. The description of the findings and of the claims that can be made based on these findings is now clear and transparent. Below are minor points that could be further clarified.

Line 331. The subtitle 'Good but not poor performers have increased c-Fos expression upon novelty exposure' does not capture the finding that the failure to find an increased c-Fos expression in poor performers can be explained by the already high expression in FE. The subtitle should be rephrased so that it is not misinterpreted as implying that c-Fos expression was consistently low in poor performers.

The subtitle as been modified to “c-Fos expression in the good and poor performers”, to avoid this potential misinterpretation (Line 331).

Line 368-371. '... would mimic the pattern of the MEC map ...' could be misinterpreted as implying that the stimulation reproduces the original MEC map. What is perhaps meant here is that the stimulation mimics that the MEC map is stable, which could be more directly stated.

The sentence in question has been modified to avoid this potential misinterpretation. The new sentence says, “In contrast, imposed stimulation of MEC neurons at consistent locations (consistent stimulation, CS), specifically, at the cues prior to the reward or immediately before the reward, would mimic the MEC map consistency and, therefore, be less disruptive to spatial memory.” (Line 368-371)

Figure S5, panels a-d. It is not clear what the comparison group is for the 'Percentage' labels on the vertical axis. What does 1005 correspond to. For example, in a, is it all recorded cells, all grid cells that were identified on a single recording day?

The figure legend has been edited to clarify that the percentage of all persistent cells is being measured.

Reviewer #3 (Remarks to the Author):

The authors have performed extensive work in this revision and I have no further concerns. This represents an important set of experiments showing how long-term grid cell stability may relate to navigation.